# Recovery Processes in a Large Offshore Wind Farm

Tanvi Gupta[1] and Somnath Baidya Roy[1]

[1]Centre for Atmospheric Sciences, Indian Institute of Technology Delhi, New Delhi, 110016, India

*Correspondence to*:  Tanvi Gupta (tanvi.gupta.iitd@gmail.com) and Somnath Baidya Roy (drsbr@iitd.ac.in)

**Abstract.** Wind turbines in a wind farm extract energy from the atmospheric flow and convert it into electricity, resulting in a localized momentum deficit in the wake that reduces energy availability for downwind turbines. Atmospheric momentum convergence from above, below and sides into the wakes replenish the lost momentum, at least partially, so that turbines deep inside a wind farm can continue to function. In this study, we explore recovery processes in a hypothetical offshore wind farm with particular emphasis on comparing the spatial patterns and magnitudes of horizontal and vertical recovery processes and understanding the role of mesoscale processes in momentum recovery in wind farms. For this purpose, we use the Weather Research and Forecasting (WRF) model, a state-of-the-art mesoscale model equipped with a wind turbine parameterization, to simulate a hypothetical large offshore wind farm with different wind turbine spacings under realistic initial and boundary conditions. Different inter-turbine spacings range from a densely packed wind farm (Case I: low inter-turbine distance of 0.5 km ~ 5 rotor diameter) to a sparsely packed wind farm (Case III: high inter-turbine distance of 2 km ~ 20 rotor diameter). In this study, apart from the inter-turbine spacings, we also explored the role of different ranges of background wind speeds over which the wind turbines operate, ranging from low wind speed range of $3 - 11.75$ ms$^{-1}$ (Case A) to that of the high wind speed range of $11 - 18$ ms$^{-1}$ (Case C). Results show that vertical turbulent transport of momentum from aloft is the main contributor to recovery in wind farms except in cases with high wind speed range and sparsely packed wind farm where horizontal advective momentum transport can also contribute equally. Vertical recovery shows a systematic dependence on wind speed and wind farm density that is quantified using low-order empirical equations. Wind farms significantly alter the mesoscale flow patterns, especially for densely packed wind farms under high wind speed conditions. In these cases, the mesoscale circulations created by the wind farms can transport high momentum air from aloft into the atmospheric boundary layer (ABL) and thus aid in recovery in wind farms. To the best of our knowledge, this is one of the first studies to look at wind farm replenishment processes under realistic meteorological conditions including the role of mesoscale processes. Overall, this study advances our understanding of recovery processes in wind farms and wind farm-ABL interactions.

## 1 Introduction

Wind power is one of the most actively growing renewable energy sources around the world with increasing emphasis on offshore wind (IRENA, 2019). A wind turbine harvests kinetic energy from the wind to produce electricity. They act as a forcing that creates a perturbation in the ABL flow. The perturbation is in the form of momentum loss and turbulent kinetic

energy increase (Baidya Roy et al., 2004; Baidya Roy, 2011; Fitch et. al, 2012). This perturbation triggers convergence of momentum from outside the wakes through turbulent and mesoscale processes that partially replenish the lost momentum so that turbines deep inside a wind farm can continue to function (Cal et al., 2010; Calaf et al., 2010; Meyers and Meneveau, 2011; Akbar and Porté-Agel, 2014; VerHulst and Meneveau, 2014; Cortina et al., 2016; Allaerts and Meyers, 2017; Cortina
et al., 2020).

A number of studies have quantitatively analyzed recovery processes for onshore wind farms using simulations from Large-Eddy Simulation (LES) models. They show that for very large wind turbine arrays, the recovery occurs mostly due to vertical momentum transport by turbulent eddies (Akbar and Porté-Agel, 2013; Cal et al., 2010; Calaf et al., 2010; Cortina et al., 2016)
whereas, for isolated turbines, recovery is dominated by horizontal advective momentum transport by the mean flow (Cortina et al., 2016). The importance of vertical transport is also confirmed by LES (Calaf et al., 2010) and wind tunnel (Cal et al., 2010) experiments that find the vertical fluxes of kinetic energy to be of the same order as that of the power extracted by the wind turbines. All existing studies in this area have used LES models with relatively small simulation domains driven by periodic, no/free-slip and other idealized boundary conditions. This prevents realistic interactions between the Atmospheric
Boundary Layer (ABL) and the free atmosphere and ignores mesoscale and larger scale processes that may contribute to the recovery in the real world.

The primary goal of this paper is to study recovery processes in offshore wind farms using numerical experiments with the WRF model, a state-of-the-art mesoscale model equipped with a wind turbine parameterization. In particular, we want to (i)
comparatively explore the spatial patterns and magnitudes of horizontal and vertical recovery processes and (ii) understand the role of mesoscale processes in momentum recovery in wind farms. Unlike earlier studies using LES models, the mesoscale model domain is large and is driven by boundary conditions from observed data. This allows us to simulate a wide range of phenomena, from turbulent to mesoscale and synoptic scale, including entrainment from the free atmosphere into the ABL. To the best of our knowledge, this is the first study to explore the contribution of both turbulent and mesoscale processes in
momentum recovery in a wind farm under realistic conditions.

Our secondary goal is to explore the role of horizontal Turbulent Kinetic Energy (TKE) advection in recovery processes. The WRF model allows users to activate the horizontal advection of TKE. Siedersleben et al. (2020) found that doing so leads to significant underestimation of wind farm impacts on TKE. However, Archer et al. (2020) showed that the underestimation was
because of a bug in the WRF code. This issue has been rectified in the latest WRF release. In this paper we have used the bug-free version to conduct sensitivity studies to evaluate the role of horizontal TKE advection in the wind farm recovery.

## 2 Methodology

### 2.1 Model description

The numerical experiments are conducted using the Weather Research and Forecasting (WRF) model (Version 4.2.1), a state-of-the-art mesoscale model that has been used for a wide range of applications. WRF solves the fully compressible, Eulerian, non-hydrostatic conservation equations for velocity, mass, energy and scalars. The vertical coordinate system is a terrain-following dry hydrostatic pressure coordinate system. The vertical grid is stretched with high resolution near the surface and coarse resolution aloft thereby allowing for better spatial resolution of ABL processes. The horizontal coordinate system uses a staggered Cartesian grid. WRF has numerous schemes to represent microphysics, cumulus convection, atmospheric radiative transfer, and ABL dynamics. The system is integrated using a Runge–Kutta 3$^{rd}$ order scheme. The model is capable of 3D variational data assimilation where boundary conditions are obtained from real world meteorological reanalyses data. Further details about the WRF model are available from Skamarock et al. (2019).

WRF is equipped with a subgrid-scale wind turbine parameterization that is capable of simulating the interactions between wind turbines and the atmosphere (Fitch et al., 2012; Fitch, 2016). In this parameterization, a wind turbine is treated as an elevated sink of momentum and source of turbulence, a concept first proposed by Baidya Roy et al. (2004). For this purpose, a sink term is added to the horizontal momentum equations and a source term is added to the Turbulent Kinetic Energy (TKE) equation. The magnitude of the source and sink terms are calculated using the thrust and power coefficients of the wind turbine.

### 2.2 Model configuration

The model simulation domain is a 1500 km × 1500 km box located deep in Arabian Sea, off the west coast of India (Fig. 1). It is discretized with 1 km grid spacing in the horizontal. The domain is intentionally kept larger than usual to ensure that wind farm wakes do not get reflected from the lateral boundaries. Preliminary sensitivity studies with smaller domain sizes starting from 300 km × 300 km showed some wake reflections but there were no boundary effects were observed with this large domain of 1500 km × 1500 km. The fine resolution domain is nested within two coarser domains of size 3000 km × 3000 km and 6309 km × 6309 km that are discretized with 3 km and 9 km grid spacing, respectively. The domain goes up to 100 hPa in the vertical and is discretized with 61 levels using a stretched grid that has 7 levels within the lowest 150 m to better resolve wind turbine-ABL interactions.

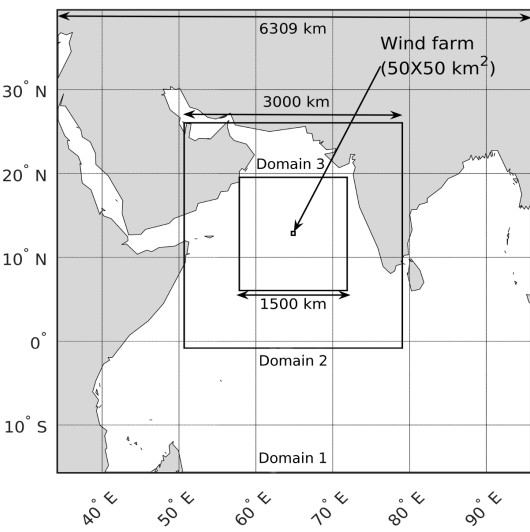

**Figure 1: Model domain showing the three nested grids. The small rectangle in the centre shows the wind farm.**

The atmospheric initial conditions, lateral boundary conditions and Sea Surface Temperature (SST) were obtained from the National Centers for Environmental Prediction Final Operational Global Analyses dataset (NCEP, 2015). Table 1 shows the

5 physics schemes used in simulations. It is to be noted that the system is closed with the 1.5-order MYNN boundary-layer scheme where TKE is prognosed while other second-order moments are parameterized. The source term in the wind farm parameterization is added to the TKE prognostic equation.

**Table 1: Physics settings used in simulations**

| Physics Settings used in WRF | Scheme Chosen |
| --- | --- |
| Microphysical processes | WRF single moment 6-class scheme, WSM6 (Hong and Lim, 2006) |
| Radiative transfer for shortwave | Goddard shortwave scheme (Chou and Suarez, 1999; Chou et al., 2001) |
| Radiative transfer for longwave | Rapid Radiative Transfer Model for General Circulations Models Scheme, RRTMG (Iacono et al., 2008) |
| Cumulus convection | New Eta Kain–Fritsch scheme (Only in domain 1) (Kain, 2004) |
| Boundary-layer scheme | 1.5-order Mellor–Yamada–Nakanishi–Niino (MYNN) (Nakanishi and Niino, 2009) |

10 For the numerical experiments, a hypothetical wind farm covering a 50 km × 50 km area is placed in the centre of the finest grid domain. The wind farm consists of wind turbines with 84 m hub-heights, 112 m rotor diameter (D), and rated capacity of 3.075 MW. The power curve of this turbine is shown in Fig. 2b. The turbines are distributed uniformly within the wind farm (Fig. 2a) with the following three different inter-turbine spacings: (I) 0.5 km (~5D) where 4 turbines are placed in each grid

cell; (II) 1 km (~10D) where 1 turbine is placed in each grid cell; and (III) 2km (~20D) where 1 turbine is placed in each alternate grid cell.

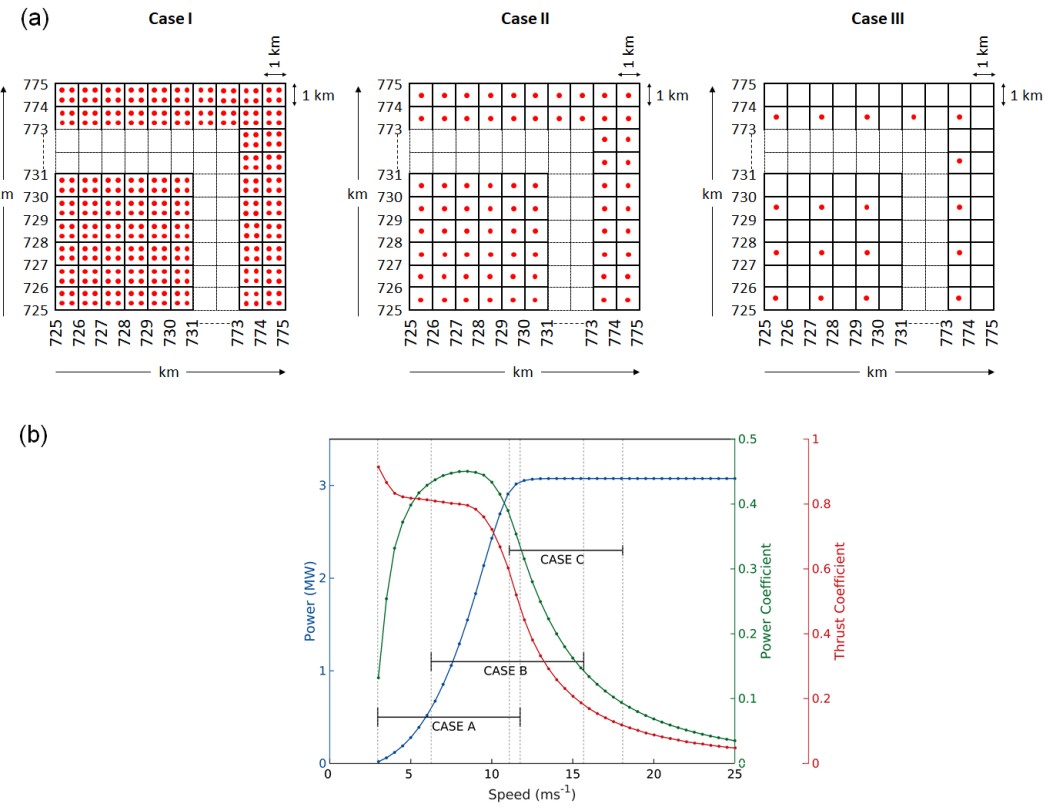

**Figure 2: (a) Wind farm layouts for Case I, Case II and Case III with inter-turbine distance of 0.5 km, 1 km and 2 km where the red dots represent the wind turbines. (b) Power curve (blue), power coefficient (green) and thrust coefficient (red) of 3.075 MW Vestas turbine as a function of wind speed. The horizontal bars represent the wind speed ranges during the 3 cases (A, B and C) simulated.**

Based on the power curve, three time periods are chosen for our simulations to cover the whole range of wind speed at which the wind turbine is operational. The time periods are: (A) Jan 10 to 13, 2015 when the wind speed is in the range 2.9 to 11.75 ms$^{-1}$; (B) June 10 to 13, 2015 when the wind speed is in the range 6.2 to 15.7 ms$^{-1}$; and (C) July 10 to 13, 2015 when the wind speed is in the range 11 to 18 ms$^{-1}$ (Fig. 2). The first days of simulations are considered as spin-up and model outputs for the remaining two days are analyzed for this study. In addition to the wind farm simulations (WF), a control simulation (CTRL) where the wind turbine parameterization is switched off is also conducted for each of these time periods. These add up to 12 simulations including 3 CTRL simulations for 3 time periods and 9 experimental WF simulations for 3 inter-turbine spacings for 3 time periods.

The above 12 simulations are conducted using the latest bug-free WRF model (Archer et al., 2020) with the TKE horizontal advection switch turned 'on'. This allows us to simulate transport of TKE by both subgrid turbulent and grid-resolved mean flow. As discussed in the introduction, we conducted sensitivity studies to explore the effect of horizontal TKE advection on replenishment. For this purpose, we repeated all the 12 simulations with the TKE advection switch turned 'off'. In this case,
TKE transport occurs only through vertical subgrid-scale transport processes.

## 2.3 Quantification of fluxes, recovery and momentum loss rate

### 2.3.1 Synoptic, mesoscale and turbulent fluxes

The synoptic and mesoscale fluxes are calculated using Avissar and Chen (1993). They posit that atmospheric variables can
be partitioned into synoptic, mesoscale and microscale components. This approach has been used to study mesoscale circulations driven by landscape heterogeneity (e.g., Li et al., 2011; Noppel and Fiedler, 2002; Zeng and Pielke, 1995). As per their formulation, a variable $\phi$ at the model grid point $(k, i, j)$ resolved by the model grid consists of a mesoscale perturbation $\phi'$ superimposed on a synoptic scale mean $\overline{\phi}$ and can be written as:

$$\phi(k,i,j) = \overline{\phi(k)} + \phi'(k,i,j),$$  (1)

where the overbar represents a horizontal average over the entire domain 3 from WRF simulations and $k, i, j$ represents the model grid points in $in\ z, x\ and\ y$ directions, respectively. Hence, the vertical synoptic scale kinematic flux of the zonal momentum (u) and meridional momentum (v) are given by:

$$\overline{UW_{synop}} = \overline{u}\,\overline{w},$$  (2)

$$\overline{VW_{synop}} = \overline{v}\,\overline{w},$$  (3)

respectively, and the vertical mesoscale kinematic flux of the zonal momentum (u) and the meridional momentum (v) are given by:

$$UW_{meso} = (u - \overline{u})(w - \overline{w})$$  (4)

$$VW_{meso} = \left(v - \overline{v}\right)\left(w - \overline{w}\right)$$  (5)

respectively. Here $u(k,i,j)$, $v(k,i,j)$ and $w(k,i,j)$ are grid resolved zonal, meridional and vertical velocities,
respectively. The microscale kinematic flux of the zonal and meridional momentums are $u'w'(k,i,j)$ and $v'w'(k,i,j)$, which are calculated by the MYNN scheme.

As per the method by Avissar and Chen (1993), the mesoscale and microscale fluxes can be estimated at each model grid cell. However, the synoptic-scale fluxes can only be estimated at each vertical level because by definition they are horizontally averaged quantities. Hence, in this study, all fluxes are horizontally averaged so they can be compared to each other.

**2.3.2 Horizontal and vertical recovery**

Recovery terms are calculated from the kinematic advective and turbulent transport terms in the Reynolds-averaged Navier–Stokes' equation (Stull, 2012) that give the contribution of advection and turbulence, respectively, to the momentum tendency. Horizontal recovery in the wind farms is calculated by taking the difference in advective momentum transport between the WF and CTRL cases as follows:

$$\text{Horizontal Recovery} = \left[ \left( -u_h \frac{\partial u_h}{\partial x} - v_h \frac{\partial u_h}{\partial y} \right) \hat{i} + \left( -u_h \frac{\partial v_h}{\partial x} - v_h \frac{\partial v_h}{\partial y} \right) \hat{j} \right]_{WF} - \left[ \left( -u_h \frac{\partial u_h}{\partial x} - v_h \frac{\partial u_h}{\partial y} \right) \hat{i} + \left( -u_h \frac{\partial v_h}{\partial x} - v_h \frac{\partial v_h}{\partial y} \right) \hat{j} \right]_{CTRL}, \quad (6)$$

where, $u_h$ and $v_h$ are the hub-height wind speeds in the zonal (x) and meridional (y) directions, and $\hat{i}$ and $\hat{j}$ are unit vectors in the zonal (x) and meridional (y) directions, respectively. The gradients are calculated using central finite differencing method. For example, the advection of zonal wind $u_h$ at a grid point (i,j) is calculated as:

$$\left( -u_h \frac{\partial u_h}{\partial x} - v_h \frac{\partial u_h}{\partial y} \right)\Bigg|_{(i,j)} = -u_h(i,j) \frac{u_h(i+1,j) - u_h(i-1,j)}{2\Delta x} - v_h(i,j) \frac{u_h(i,j+1) - u_h(i,j-1)}{2\Delta y}, \quad (7)$$

The advection of meridional wind ($v_h$) is calculated in a similar way. The vector difference in Eq. (6) is projected against the prominent wind direction from the CTRL case.

prominent wind direction from the CTRL case.

Vertical recovery is calculated by taking the difference in vertical turbulent transport between the WF and the CTRL cases as follows:

$$\text{Vertical Recovery} = \left[ -\left( \frac{\partial u'w'}{\partial z} \right) \hat{i} - \left( \frac{\partial v'w'}{\partial z} \right) \hat{j} \right]_{WF} - \left[ -\left( \frac{\partial u'w'}{\partial z} \right) \hat{i} - \left( \frac{\partial v'w'}{\partial z} \right) \hat{j} \right]_{CTRL}, \quad (8)$$

where $u'w'$ and $v'w'$ are the vertical kinematic turbulent flux of the zonal and meridional momentums, respectively. These fluxes are calculated by the MYNN scheme at the model grid points. They are interpolated to the heights of 28 m ($z_1$) and 140 m ($z_2$), the lower and upper wind turbine blade tip heights, respectively. The vertical gradient of the fluxes are then calculated between these two levels. For example, the vertical turbulent transport of zonal momentum is calculated as:

$$\frac{\partial u'w'}{\partial z} = \frac{u'w'|_{z_2} - u'w'|_{z_1}}{z_2 - z_1}, \quad (9)$$

Vertical turbulent transport of meridional momentum is calculated in a similar way. In this approach we ignore the vertical transport of momentum by the large-scale vertical wind. This term is very small compared to the turbulent transport because large-scale vertical motion is typically close to zero in the ABL. The vector difference in Eq. (8) is projected against the prominent wind direction from the CTRL case.

### 2.3.3 Momentum loss rate

The momentum loss rate (ms$^{-2}$) is calculated as per Fitch et al. (2012):

$$\frac{\partial |V|}{\partial t}\bigg|_{(k,i,j)} = -\frac{\frac{1}{2} N_t(i,j) C_T |V(k,i,j)|^2 A(k,i,j)}{z(k+1,i,j) - z(k,i,j)},$$

(10)

where, $V(k,i,j) = [u(k,i,j), v(k,i,j)]$ is the horizontal wind velocity (ms$^{-1}$), $N_t(i,j)$ = number of turbines per square

metre (m$^{-2}$) for the grid cell (i, j), $C_T$ = turbine thrust coefficient which is dependent on $V$ at hub-height , $A(k,i,j)$ = cross-sectional rotor area (m$^2$) and $z(k,i,j)$ = height (m) of level k. The momentum loss rate is projected along the prominent wind direction from the CTRL case for it to be consistent with the recovery terms.

### 2.4 Stability Calculations

We estimate the environmental stability of the experimental domains using two different approaches. In the first approach, we

calculate the Flux Richardson number (R$_f$, Stull, 2012) as per Eq. (11).

$$R_f = \frac{\left(\frac{g}{\theta_v}\right)(\overline{w'\theta_v'})}{(\overline{u'w'})\frac{\partial u}{\partial z} + (\overline{v'w'})\frac{\partial v}{\partial z}},$$

(11)

where, g is acceleration due to gravity, $\theta_v$ is virtual potential temperature, and $\overline{w'\theta_v'}$ is vertical kinematic turbulent flux of virtual potential temperature. As per Sorbjan and Grachev (2010), R$_f$ < -0.02, -0.02 < R$_f$ < 0.02, and R$_f$ > 0.02 correspond to statically unstable, near-neutral, and stable environments, respectively. Moreover, R$_f$ < 1 indicates a dynamically unstable

environment (Stull, 2012).

In the second approach, we used the non-local definition of static stability from Stull (2012) based on the virtual potential temperature lapse rate ($-\frac{\partial \theta_v}{\partial z}$) of the ABL of the wind farm cases. A positive, zero, and negative value of the lapse rate indicates statically unstable, neutral, and stable conditions, respectively. However, if a neutral layer has an underlying unstable layer,

then the entire column is considered unstable.

## 3. Results

### 3.1 Power generation in the wind farms

The wind regime and the power generated by the wind farm experiments are shown in Fig. 3. Figure 3a shows the wind rose for CTRL case wind speeds over the wind farm area. For case A, 50% of winds are north-easterly. In contrast, the winds are mostly west-southwesterly for case B (84%) and C (100%). As discussed earlier, case A has the weakest wind speed while case C has the strongest wind speed.

**Table 2: Power generated (GW) by the wind farms. Wind farm efficiencies are given in parentheses.**

| Cases | I | II | III |
|-------|-----|-----|-----|
| A | 2.81 (31%) | 1.60 (70%) | 0.51 (90%) |
| B | 8.68 (31%) | 5.44 (78%) | 1.64 (94%) |
| C | 24.15 (81%) | 7.65 (100%) | 1.92 (100%) |

In case I, 10000 wind turbines with 3.075 MW rating are placed 0.5 km apart over a 2500 km$^2$ area leading to a 30.75 GW installed capacity. Cases II and III have 2500 and 625 turbines spaced 1 km and 2 km apart leading to 7.687 GW and 1.921 GW installed capacities, respectively. The power generated by the wind farms are shown in Table 2. As expected, increasing wind speeds increase power generation with case A generating the least amount of power and case C the most. Sparsely packed wind farms produce less power because of the lower installed capacity but their efficiencies are higher because increased inter-turbine spacing reduces the wake effects. For example, the densely packed case A-I produces 2.81 GW of power while the sparsely packed case A-III produces 0.51 GW. However, case A-I operates at 31% efficiency while case A-III at 90% efficiency. Cases C-II and C-III are 100% efficient. This is because the high wind speed in case C and the low wake effects due to the high inter-turbine spacing ensures that the wind speed is always above the rated wind speed of 12 ms$^{-1}$ of the turbine (Fig. 2b).

The spatial patterns of power generation (Fig. 3b) averaged over 48 hours show that generation is the highest for the first few rows of turbines of the wind farms at the upwind edges in the direction of the dominant winds. The power production decreases inside the wind farm due to wake effects for all cases except C-II and C-III. As explained above, the high wind speed and low wake effects result in the turbines in these two cases always operating at the rated power.

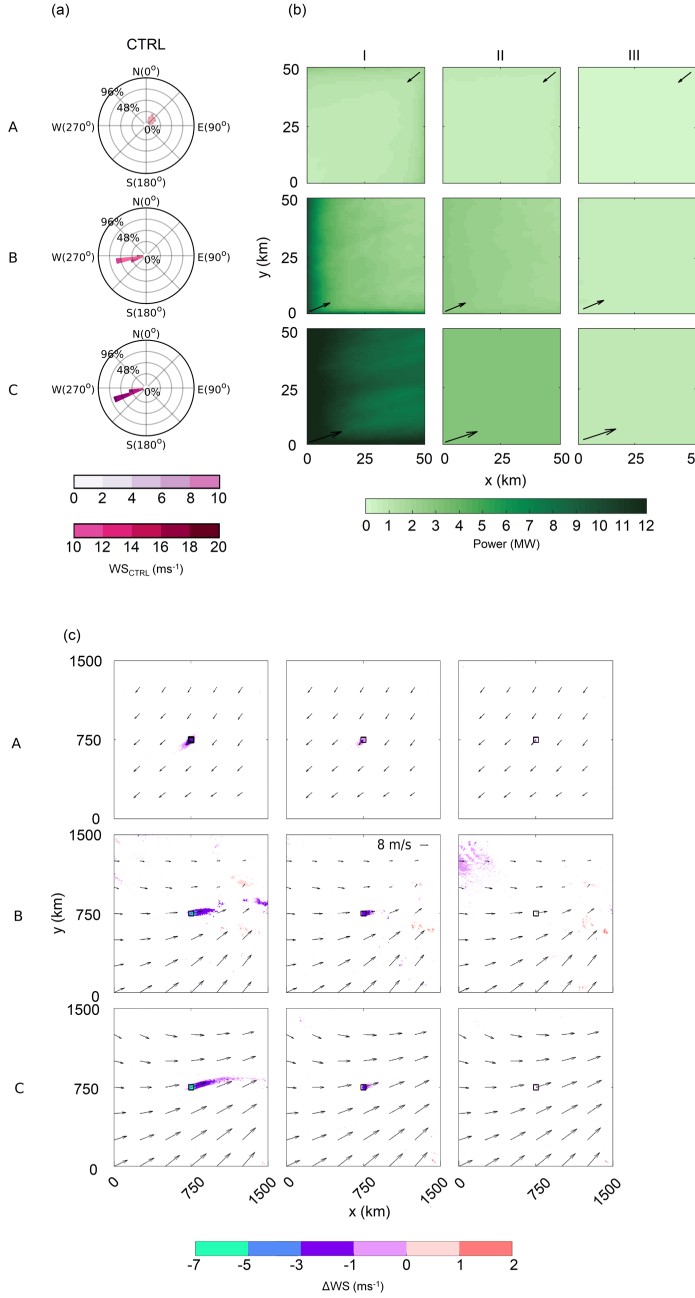

**Figure 3: (a)** Wind rose diagrams depicting hub-height wind speeds, WS (ms⁻¹) over the wind farm area for the CTRL case, **(b)** averaged power (MW) generated in the WF cases, and **(c)** averaged difference in wind speed, ΔWS (ms⁻¹) over the rotor depth (28m-140m) (WF-CTRL) for **(A)** January, **(B)** June, and **(C)** July with **(I)** 0.5 km, **(II)** 1 km and **(III)** 2 km inter-turbine spacings. In (b), the plot depicts the wind farm only, that is the black square in (c) and the black arrow shows the prominent wind direction. In (c), only the statistically significant results (p<0.01) are shown. White colored regions represent areas where the wind speed differences are not statistically significant. The vectors represent the wind in the corresponding CTRL cases, every 250th vector is shown.

## 3.2 Stability in wind farms

The stability conditions for the different wind farm cases are estimated on an hourly basis for both the approaches. $R_f$ is averaged from the surface to the wind turbine rotor tip height of 140 m of the wind farm cases comprising of roughly 7 eta levels and over the wind farm. Potential temperature lapse rate is evaluated for the entire atmospheric column averaged over the 50 km x 50 km wind farm for each hour. The stability is further classified as per methodology given in section 2.4. The percentage occurrences of different stability conditions are shown in Table 3. Results show that Case A is primarily statically unstable, Case B is a mix of statically stable and unstable, and Case C is statically stable. Moreover, all cases are dynamically unstable.

**Table 3: Percentage occurrence of different stability conditions**

| Cases | $R_f$ based dynamic stability | | | $R_f$ based static stability | | | Lapse rate based static stability | | |
|---|---|---|---|---|---|---|---|---|---|
| | Stable | Neutral | Unstable | Stable | Near-Neutral | Unstable | Stable | Near-Neutral | Unstable |
| A-I | 0% | 0% | 100% | 2.1% | 2.1% | 95.8% | 0% | 14.6% | 85.4% |
| A-II | 0% | 0% | 100% | 4.2% | 0% | 95.8% | 0% | 0% | 100% |
| A-III | 0% | 0% | 100% | 4.2% | 2.1% | 93.8% | 0% | 0% | 100% |
| B-I | 0% | 0% | 100% | 47.4% | 26.3% | 26.3% | 78.4% | 16.2% | 5.4% |
| B-II | 0% | 0% | 100% | 55.8% | 20.9% | 23.3% | 40.6% | 18.8% | 40.6% |
| B-III | 0% | 0% | 100% | 50.0% | 36.4% | 13.6% | 50.0% | 6.3% | 43.8% |
| C-I | 0% | 0% | 100% | 100% | 0% | 0% | 97.9% | 2.1% | 0% |
| C-II | 0% | 0% | 100% | 100% | 0% | 0% | 87.5% | 5.0% | 7.5% |
| C-III | 0% | 0% | 100% | 100% | 0% | 0% | 76.3% | 18.4% | 5.3% |

## 3.3 Wind farm wakes

A comparison of the wakes from wind farms for the different experiments are shown in Fig. 3c. Wind farm wakes are regions with weak wind speeds downwind of a wind farm. In our study, we define wakes as regions where the differences in wind speeds averaged over the rotor depth (28 m – 140 m) and in time (48 hours) between the experiment and control cases (WF-CTRL) is negative and statistically significant at $p < 0.01$ (Fig. 3c). Statistical significance is estimated using the Wilcoxon Sign Rank Test  (Wilcoxon, 1945) that is a well-known, non-parametric paired difference test. The test was chosen because the input data was not following the normal distribution. It is a non-parametric test that does not require the data to follow a normal or other known distribution. Hence, this is a common alternative to other tests that need the data to follow a normal distribution.

The test is conducted for each of the 1500 x1500 points in the domain 3 of simulations. For each point, there are 48 hourly wind speed difference (WF-CTRL) values, and we test for the null hypothesis that the differences come from a population with zero median. The results shown in Fig. 3c are for the points in the domain where we reject the null hypothesis and claim that the difference in the wind speeds (WF-CTRL) is not because of random chance at a confidence level greater than 99%

($p<0.01$).

The wake lengths are calculated as distance from the wind farm downwind edge to the grid cell where the difference becomes non-significant. Several interesting features of the wakes can be observed from this figure. First, the wakes follow the predominant wind directions resulting in wakes towards the southwest for case A and east-northeast in cases B and C. Second,

the wake length increases with the increasing wind speeds. For example, the wake length increases from 80 km to 223 km and further to 521 km with the increasing wind speed for cases A-I, B-I and C-I, respectively. The same pattern is observed for cases II. It is important to note that for case A, the atmosphere is predominantly statically unstable, case B is mix of stable and unstable, and C is stable. Many studies (Djath et al., 2018; Platis et al., 2018; Cañadillas et al., 2020; Platis et al., 2020) have observed smaller wakes in unstable atmosphere as compared to stable. Therefore, the smaller wake length for Case A is likely

to be because of both the unstable environment and low wind speeds. Third, the wake length decreases with increasing inter-turbine spacing. For example, the wind farm wake length decreases from 224 km to 85 km and further to 0 km for cases B-I, B-II and B-III, respectively. A similar pattern is observed for the other cases. Fourth, almost no discernible wind farm wake is observed for sparsely packed wind farms with 2 km spacing in case III.

The wakes are extremely strong and long in B-I and C-I. In these cases, the wind speeds are strong, the atmosphere is relatively stable, and the wind farms are densely packed. For example, in C-I, the wake is ~521 km long. Here, an average wind speed reduction of 7% at hub-height can be seen with a maximum hub-height wind speed reduction of 31% behind the wind farm in the direction of the wind flow. This reduction is comparable with Platis et al. (2018) and Platis et al. (2020) who found a maximum deceleration of 40% and 43% in the wake of offshore wind farms, respectively. The average hub-height wind speed

reduction is 15% in regions immediate downwind (50 km) of the wind farm. This reduction is comparable with Fitch et al. (2012) who reported a 11% decay 11 km behind a $10 \times 10$ km modeled wind farm. A similar wind speed reduction of 8-9% was also observed by Christiansen and Hasager (2005) immediately downstream of the Horns Rev and Nysted offshore wind farms.

### 3.4 Circulation patterns around offshore wind farms

Figure 4a shows the difference (WF-CTRL) in horizontal wind velocity component on a vertical cross-section along the prominent wind direction. The cross-sections are averaged over a 30 km wide band around a line running through the center of the wind farm. Figure 4b shows the same but for vertical velocity. For clarity, we focus on a region 100 km up and downwind of the wind farms depicted by the dashed box instead of the entire 1500 km span of the domain. One important thing to note

here is than in case A, unlike cases B and C, a positive value in the difference plot indicates a velocity deficit because the prominent wind is towards the negative x-direction. Four major features are observed from these plots. First, there is a strong reduction in the horizontal wind speed up to 7 ms⁻¹ within the wind farm due to extraction of kinetic energy by the wind turbines. Second, the wind farm wakes are clearly visible as strong wind speed deficits up to 5 ms⁻¹ downwind of the wind farms. The wind farm wakes can spread vertically up to the top of the ABL at around 1–1.5 km. These effects are stronger for higher wind speeds and higher wind farm density.

The third feature is the reduction in wind speed upwind of the wind farms while the fourth is the increase in horizontal wind speed above the wind farm that are visible in all the simulations. These two features occur because the wind farm acts as a barrier to the flow. The incoming wind slows down as it approaches the wind farm causing the observed upwind stilling. A part of the incoming flow rises above the wind farm at the upwind edge. The vertical displacement is quite deep, even going much higher than the ABL top. The lifted flow then descends near the downwind edge of the wind farm. This pattern is stronger for higher wind speeds and higher wind farm density. Similar patterns of upwind stilling and lifting have also been observed by other studies (Bleeg et al., 2018; Porté-Agel et al., 2020).

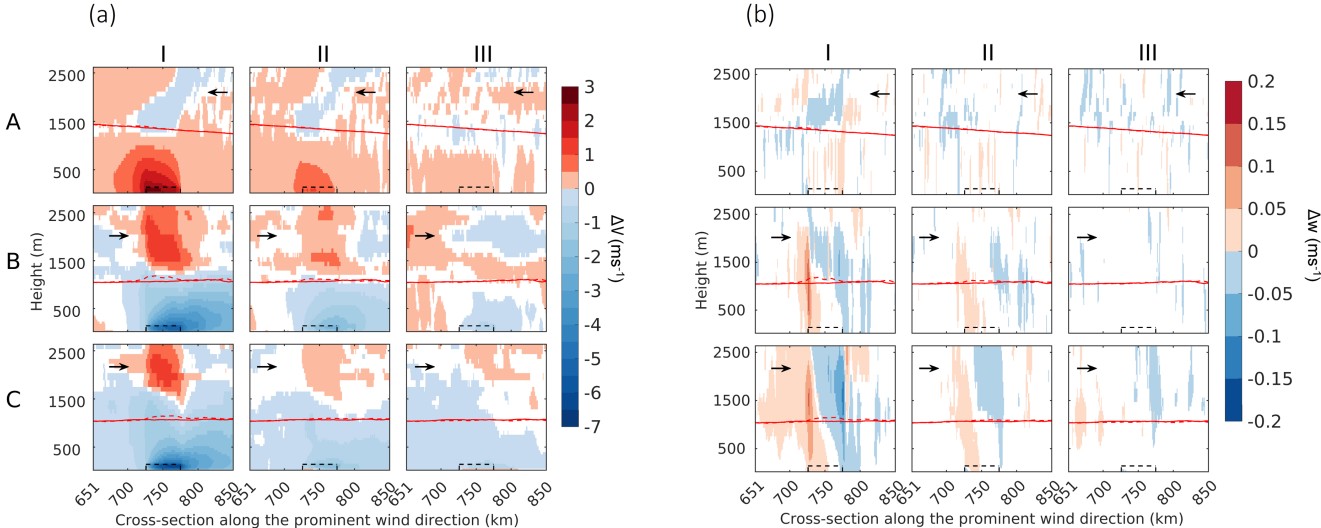

**Figure 4: (a) Difference in horizontal wind velocities, ΔV (ms⁻¹) (WF-CTRL) and, (b) Difference in vertical wind velocity, Δw (ms⁻¹) (WF-CTRL) on a vertical cross-section along the predominant wind direction for case: (A) January, (B) June, and (C) July with (I) 0.5 km, (II) 1 km, and (III) 2 km turbine spacings. Only the statistically significant results (p<0.01) are shown here. The white colored regions represent areas where the differences are not significant. The black dashed box depicts the wind farm cross-section. The red dashed line depicts ABL height (WF case) and red solid line depicts ABL height (CTRL case). The arrows represent the predominant wind direction.**

Corroborating evidence of the lifting is visible in the vertical cross-section of vertical wind velocity difference, especially for cases A, B and C, I and II (Fig. 4b). These plots show an upward motion at the upwind edge of the wind farm and a downward motion beyond that. There is no statistically significant signal in vertical velocity for the sparsely packed wind farms in cases

III. The lack of a strong signal in the vertical velocity field is not unusual because vertical velocities in mesoscale flow are notoriously hard to simulate (Weaver, 2009).

## 3.5 Effect of wind farms on synoptic, meso and micro-scale momentum fluxes

Figure 5 compares the vertical profiles of the synoptic, meso and micro-scale vertical momentum fluxes for the WF cases with the CTRL case. The synoptic (large scale flow>1000 km), meso (1- 1,000 kilometers) and micro-scale fluxes (<1 km) are calculated as per the equations given in section 2.3.1. The figure shows the difference (WF-CTRL) in the flux profiles between the experimental and the corresponding control cases horizontally averaged over the wind farm area for the 2-day simulation period. Results show that the difference in synoptic scale flux profiles is negligible and hence, not visible in the figure. This

indicates that the wind farms do not affect synoptic scale flow and momentum transport in a significant way.

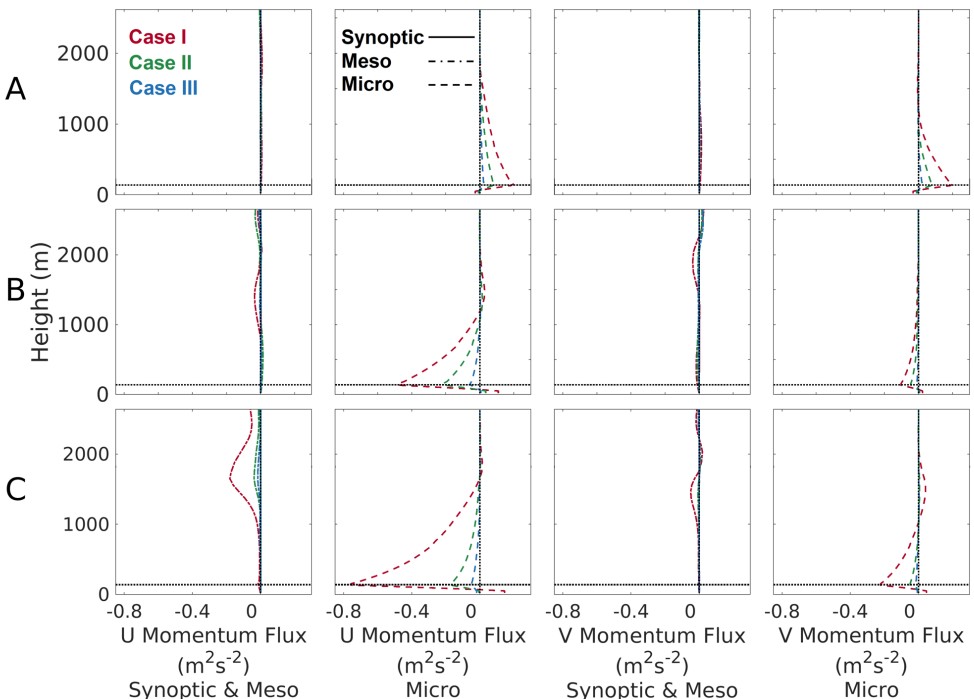

**Figure 5: Difference (WF-CTRL) in vertical profiles of vertical synoptic (solid line), mesoscale (dashed dotted line), and microscale fluxes (dashed line) averaged over the wind farm for the simulation period for (A) January, (B) June and (C) July. Horizontal black dotted line shows the height of the upper tip of the wind turbine rotor.**

Wind farms affect vertical mesoscale momentum fluxes only in case C-I. In this case, the difference in flux is negative throughout the lower atmosphere with a minimum of -0.155 $m^2s^{-2}$ at a height of 1650 m that is much above the ABL height of ~1000 m. This pattern indicates that the mesoscale flow generated by the wind farm leads to a downward transport of momentum from the free atmosphere into the ABL. Because the predominant wind direction is towards the east, the signal is

observed only in the u-momentum flux but not in the v-momentum flux. The magnitude of the downward mesoscale momentum transport is smaller than but of the same order of magnitude as the microscale transport term.

The other cases do not show any significant mesoscale transport even though they create mesoscale perturbations in the flow (Fig. 4). This is because, the mesoscale fluxes associated with the updrafts and downdrafts are confined to narrow bands that disappear upon spatial averaging. Hence, the mesoscale perturbation in these cases do not lead to a strong signal in vertical mesoscale fluxes.

Wind farms strongly affect the microscale fluxes. In this figure, a positive (negative) flux difference indicates an upward (downward) transport compared to the control except for case A where the opposite is true because the horizontal wind is towards the negative x-direction. The results show that subgrid-scale turbulent eddies transport momentum to the turbine hub-height level from above and below the rotors. The magnitudes of the microscale fluxes are much larger than the mesoscale fluxes indicating that turbulent transport is the primary mode of vertical replenishment in wind farms.

The magnitudes of the fluxes increase with increasing wind speed and wind farm density. Case A with the weakest wind has the lowest fluxes while case C with the strongest wind has the highest fluxes. Case I with the densely packed wind farm has the highest flux while case III with the sparsely packed wind farm has the lowest flux. The relative magnitudes of the u and v momentum fluxes depend on their wind directions. In case A, where the winds are northeasterly with u and v components of similar magnitudes, the corresponding fluxes also have similar magnitudes. On the other hand, in cases B and C, where the winds are west-southwesterly with stronger u components than v, the u momentum flux is much higher than the v momentum flux.

We can see signs of an interesting phenomenon when comparing the mesoscale and microscale fluxes in C-I. Both fluxes are negative, indicating a downward transport of momentum. However, their spatial patterns are complementary. The mesoscale flux is negative from the free atmosphere up to the ABL height while the microscale flux is negative from the ABL top to the turbine hub-height. This implies that mesoscale processes are transporting momentum from above into the ABL and microscale fluxes are transporting momentum from the ABL top to the hub-height. Therefore, it is possible that mesoscale momentum transport aids in the wind farm recovery by making more momentum available for downward mixing by turbulence. This finding is different from Antonini and Caldeira (2021) who studied an idealized infinite wind farm and found that energy extracted by wind farms are replenished from energy within the ABL. Their finding might be an artefact of their experiment design where the infinite wind farm does not leave any space within the model domain for mesoscale circulations to develop.

## 3.6 Recovery in the wind farm

### 3.6.1 Quantitative assessment of vertical and horizontal recovery

Figure 6a and 6b show the spatial patterns of vertical and horizontal recoveries, respectively, over the wind farm. The arrows show the dominant direction of the wind flow at turbine hub-height level. It can be seen that the vertical recovery occurs all
over the wind farm, but it is the weakest at the upwind edges. The magnitude of vertical recovery is stronger for stronger wind speeds and higher wind turbine density.

The horizontal recovery has an interesting spatial pattern. For cases with high turbine density like A-I, B-I and C-I, horizontal recovery occurs mostly at the upwind side. In these cases, power production is maximum at the upwind edges (Fig. 3b) creating
a strong localized gradient in momentum. Because the horizontal wind speeds are strong, horizontal advection can replenish the momentum deficit faster than vertical turbulent diffusion. For cases with low turbine density like A-III, B-III and C-III, horizontal recovery also occurs throughout the wind farm as bands perpendicular to the predominant wind direction. In these sparsely packed cases, the wind turbines are placed in alternate grid cells. Horizontal recovery only affects the grid cells containing the wind turbines, resulting in the banded pattern with peak magnitudes similar to that of vertical recovery. Note
that in case III, the turbines are placed in every alternate grid cell to ensure a 2 km spacing. Hence, the data points in this case are averaged over 4 grid cells covering 2 km × 2km leading to a relatively smoother plot. The vertical and horizontal recovery patterns from densely packed cases C-I and B-I are similar to the findings of Cortina et al. (2020), where finite-sized high-density wind farms are simulated using an LES model under neutral stability conditions.

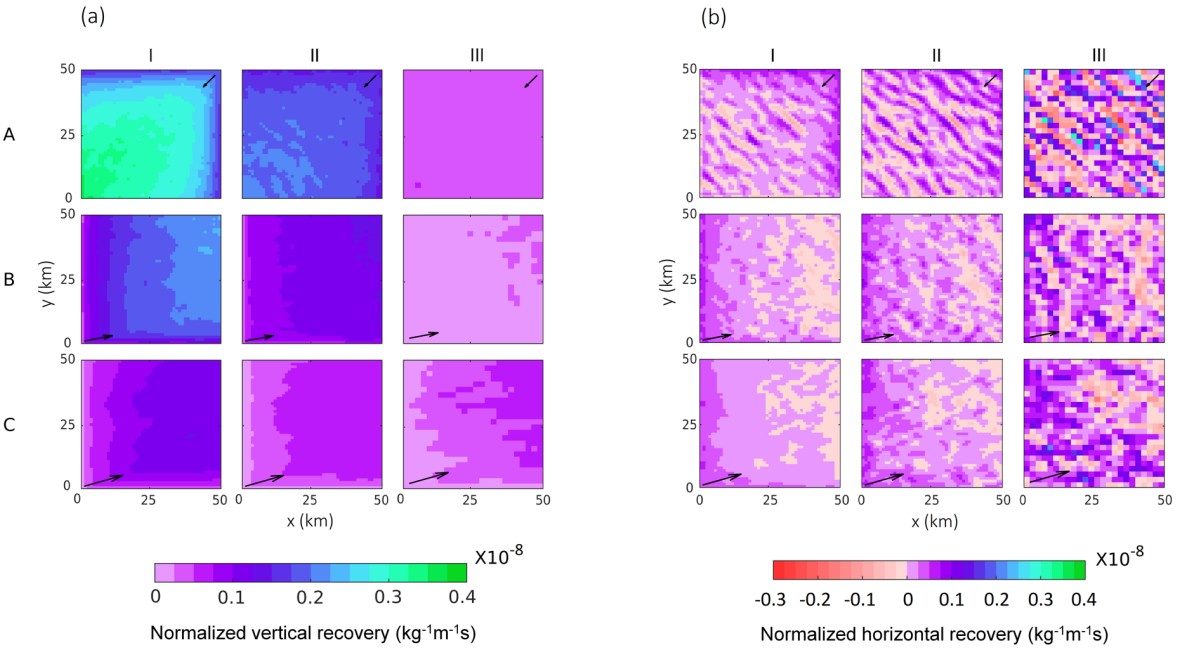

**Figure 6: (a) Vertical recovery (normalized by power) and (b) Horizontal recovery (normalized by power) for case (A) January, (B) June, and (C) July with (I) 0.5 km, (II) 1 km, and (III) 2 km turbine spacings over the wind farms. Black arrows (scaled to the magnitude of averaged wind speed) show the prominent wind direction from the CTRL case.**

Table 4 shows the averaged momentum loss rate, vertical recovery and horizontal recovery over the wind farms. The momentum loss rate over the wind farm is calculated as described in section 2.3.3. As per Eq. (10), momentum loss rate is directly proportional to three main components: (i) the number of wind turbines per square metre, (ii) the thrust coefficient of the turbines that is a function of horizontal velocity and (iii) the square of horizontal velocity. As expected, the magnitude of momentum loss rate decreases with decreasing turbine density. The relationship between momentum loss rate and wind speed

is complex. In case I, the momentum loss rate increases with increase in wind speed. However, for case II and III, the momentum loss rate first increases from case A to B and then, decreases from case B to C. In these cases, for lower wind speed range in cases A and B, the reduction in thrust coefficient is relatively small (Fig. 2). Hence, the wind speed drives the momentum loss rate and with increase in wind speed, momentum loss rate increases. However, as the wind speed further increases to the level of case C, the drop in thrust coefficient is large. In this case, the drop in thrust coefficient dominates the

momentum loss rate. Due to reduction in thrust coefficient, momentum loss rate decreases. Overall, the momentum loss rate over the wind farm is dependent on a complex interplay between the three earlier mentioned terms.

Vertical recovery follows the same pattern with respect to wind speed and inter-turbine spacing (Table 4) as momentum loss rate. This is because vertical recovery is a linear function of the momentum loss rate's magnitude (Fig. 7a). Horizontal recovery

increases with increase in wind speed and decrease in inter-turbine spacing for all the cases. As per Eq. (6), the horizontal recovery is directly proportional to (i) the wind speed upstream of the wind turbine and (ii) spatial gradient of wind speeds across a wind turbine. Therefore, as wind speed increases from case A to C, the horizontal recovery increases. Similarly, as the inter-turbine spacing increases from case I to III, the spatial gradient of wind speed decreases and hence horizontal recovery decreases.

The wind farm momentum extracted by the turbines is largely replenished by vertical recovery. The replenishment by vertical recovery is in the 47.2% to 70.8% range and it increases with decreasing turbine density similar to the findings of Cortina et al. (2017). Horizontal recovery plays a relatively small role, replenishing about 6.6% to 31.6% of momentum loss. Overall, the total replenishment of momentum in the wind farms lies in between 77.3% (case III) to 79% (case II). This indicates that

the two recovery terms considered here are not able to completely replenish the momentum loss.

**Table 4: Momentum loss rate (x $10^{-3}$), ms$^{-2}$, vertical recovery (x $10^{-3}$), ms$^{-2}$, and horizontal recovery (x $10^{-3}$), ms$^{-2}$ averaged over the wind farm and the simulation period. The numbers in the parenthesis give the percentage recovery with respect to the corresponding momentum loss rate.**

| Cases | I | II | III |
|-------|---|----|----|

|  | Mom. Loss Rate | Vert. Recovery | Horiz. Recovery | Mom. Loss Rate | Vert. Recovery | Horiz. Recovery | Mom. Loss Rate | Vert. Recovery | Horiz. Recovery |
|---|---|---|---|---|---|---|---|---|---|
| A | -3.93 | 2.78 (70.6%) | 0.26 (6.6%) | -1.62 | 1.15 (70.8%) | 0.20 (12.3%) | -0.47 | 0.32 (67.6%) | 0.04 (9.1%) |
| B | -7.61 | 5.37 (70.5%) | 0.58 (7.6%) | -3.27 | 2.22 (67.8%) | 0.35 (10.7%) | -0.86 | 0.55 (63.6%) | 0.11 (12.7%) |
| C | -12.32 | 8.39 (68.1%) | 1.17 (9.5%) | -2.59 | 1.52 (58.7%) | 0.43 (16.5%) | -0.61 | 0.29 (47.2%) | 0.19 (31.6%) |

### 3.6.2 Characterization of vertical recovery

The results above show that replenishment in wind farms is primarily due to vertical recovery. Therefore, in this section, we further analyze vertical recovery by characterizing its relationship with momentum deficit and upwind wind speeds (Fig. 7). We also develop empirical equations to quantify the relationship.

Vertical recovery varies in direct proportion to the momentum extracted by the turbines. This is evident from Fig. 7a, where vertical recovery is plotted with respect to momentum loss rate. The increase in vertical recovery is very sharp at the transition from case II to case I at momentum loss rate $\sim 3.5 \times 10^{-3}$ in Fig. 7a. There are also some points where the vertical recovery is low even though the momentum loss rate is high. These are the points on the upwind edges of the wind farm where the horizontal recovery dominates leading to lower values of vertical recovery. Figures 7b, c and d depict the scatter plot of vertical recovery with respect to upwind wind speeds for the three different wind turbine densities. These plots show that vertical recovery increases with wind speed until the rated wind speed of the wind turbine is reached. Beyond the rated wind speed, vertical recovery starts decreasing. This is because the momentum deficit increases until the rated wind speed and then decreases.

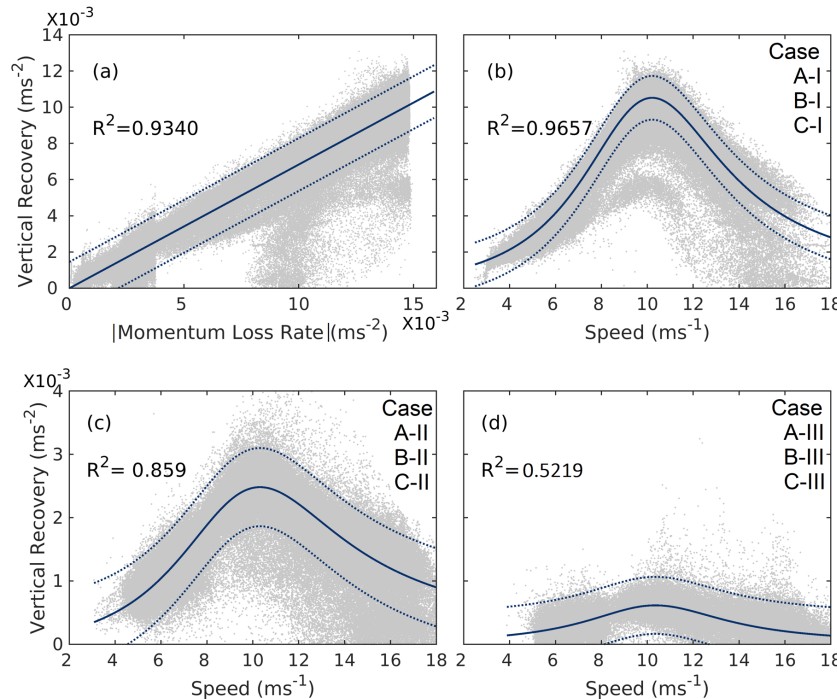

**Figure 7: Scatter plot of vertical replenishment with (a) absolute momentum loss rate for all cases; and upwind wind speed for (b) case A-I, B-I and C-I, (c) case A-II, B-II and C-II and (d) case A-III, B-III and C-III. The solid blue line indicates the best fit and dashed blue line indicates the 99% prediction intervals. The total number of data points in Fig. 7a are 9x50x50x48 and in Fig. 7b–d are 3x50x50x48.**

The empirical relationships for vertical recovery as a function of momentum loss rate and wind speed and are:

$$\text{1)} \quad y = 0.6841|x_1| - 0.0000103, \tag{12}$$

$$\text{2)} \quad y = \frac{0.009567x_2 + 0.06145}{x_2^2 - 19.35x_2 + 108.7}, \qquad \text{d=0.5 km} \tag{13}$$

$$\text{3)} \quad y = \frac{0.004154x_2 + 0.008504}{x_2^2 - 18.95x_2 + 109.8}, \qquad \text{d=1 km} \tag{14}$$

$$\text{4)} \quad y = \frac{0.0001853x_2 + 0.006901}{x_2^2 - 20.4x_2 + 118.4}, \qquad \text{d= 2 km} \tag{15}$$

Where, $y$ is vertical recovery (ms$^{-2}$), $x_1$ is momentum loss rate (ms$^{-2}$), $x_2$ is upwind hub-height wind speed (ms$^{-1}$) and d is the inter-turbine spacing.

### 3.7 Dependence of vertical recovery on mesoscale fluxes

Section 3.5 shows that only case C-I generates mesoscale fluxes of the same order of magnitude as turbulent fluxes. It can be seen in Fig. 5 that the u momentum flux is much higher than the v momentum flux for case C-I. Hence, here we have only considered the vertical mesoscale flux of u momentum. It is likely that downward transport of momentum by mesoscale eddies in this case can make more momentum available for turbulent eddies and thereby aid in recovery. To further investigate this issue, we look at the relationship between vertical recovery and the vertical mesoscale flux of u momentum (Fig. 8). The vertical mesoscale flux of the horizontal momentum is calculated as per the formulation given in section 2.3.1 (Eq. 4) and then integrated between 1000–2500 m levels. It can be seen that as the mesoscale fluxes of horizontal u momentum increases in magnitude the vertical recovery increases. This suggests a direct dependence of vertical recovery on mesoscale fluxes.

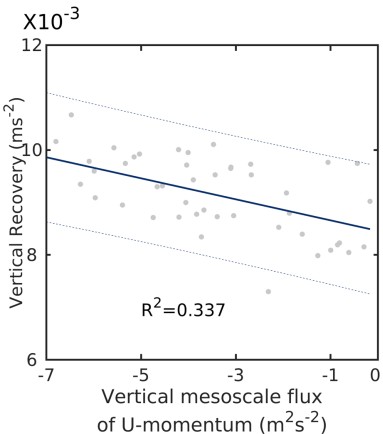

**Figure 8: Scatter plot of averaged vertical recovery (at hub-height) on averaged vertical mesoscale flux of u momentum (integrated between 1000–2500m) for case C-I. The values are averaged over the wind farm. The solid blue line shows the best-fit and the dashed blue lines show the 95% prediction intervals.**

### 3.8 Effect of stability on vertical recovery

Figure 9a–i shows a relationship between recovery and Flux Richardson number for all the cases. As discussed earlier in Section 3.2, Case A is primarily statically unstable, Case C is statically stable, and Case B is a mix of statically stable and unstable. Moreover, all cases are dynamically unstable. The points in the scatter plot are instantaneous hourly data spatially averaged over the wind farm and are segregated by wind speed. The figure shows that for a given wind speed range, there is no statistically significant correlation (Kendall, 1970) between recovery and stability, even for case B where both stable and unstable conditions are prevalent.

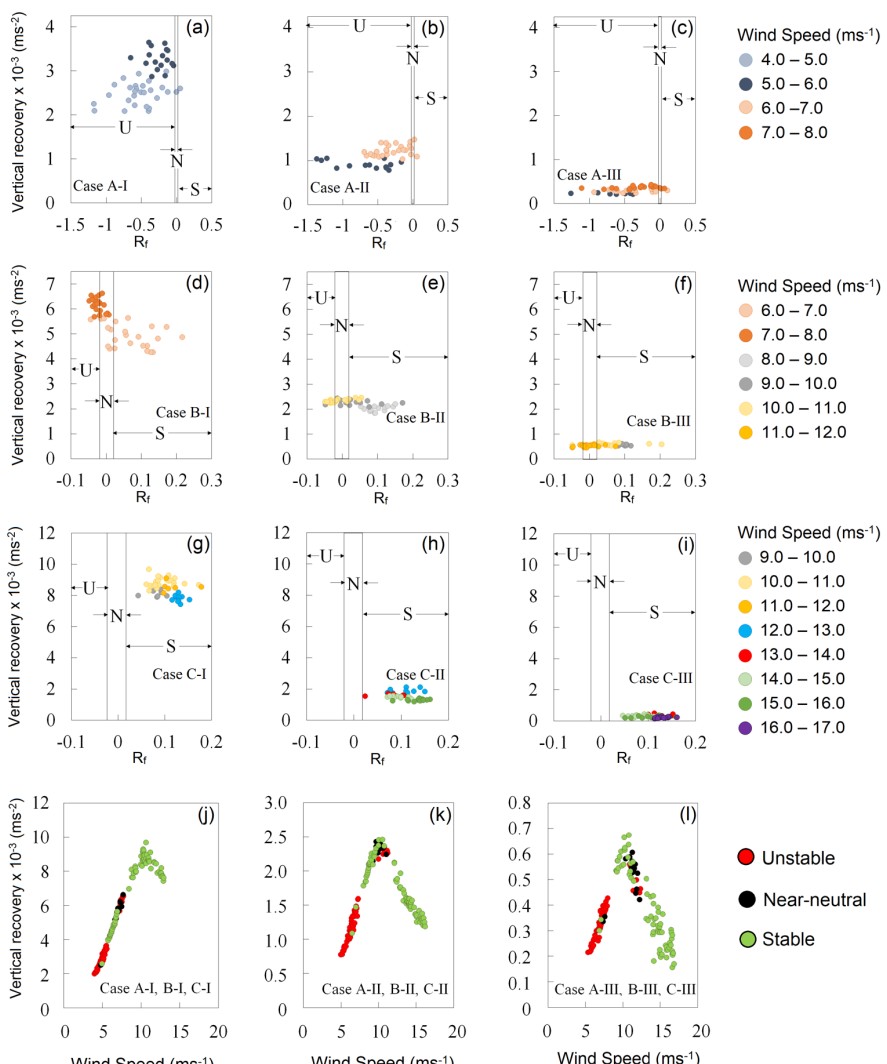

**Figure 9: (a–i)** Relationship between vertical recovery (ms$^{-2}$) and Flux Richardson number ($R_f$) for all the nine cases. The recovery data is segregated for different wind speed (WS) ranges. U, N and S depicts unstable, near-neutral and stable atmospheric conditions, respectively, **(j–l)** Same as Fig. 7b, c and d, but the vertical recovery (ms$^{-2}$) is spatially averaged over 50 km x 50 km wind farm points for each hour and data is binned as per different static stability conditions that are calculated using $R_f$. The total number of data points in Fig. 9j–l are 3x48 each.

To further explore the relationship between stability, wind speed, and recovery, we re-plotted Fig. 7b–d as Fig. 9j–l by segregating the points according to stability calculated using the $R_f$. There are fewer data points in Fig. 9j–l compared to Fig. 7b–d because we have spatially averaged the variables over the whole wind farm for each hour. This helps us eliminate the effect of spatial variation of recovery within the wind farm (Fig. 6). Figure 9j–l shows that the variation in vertical recovery has a strong relationship with speed.

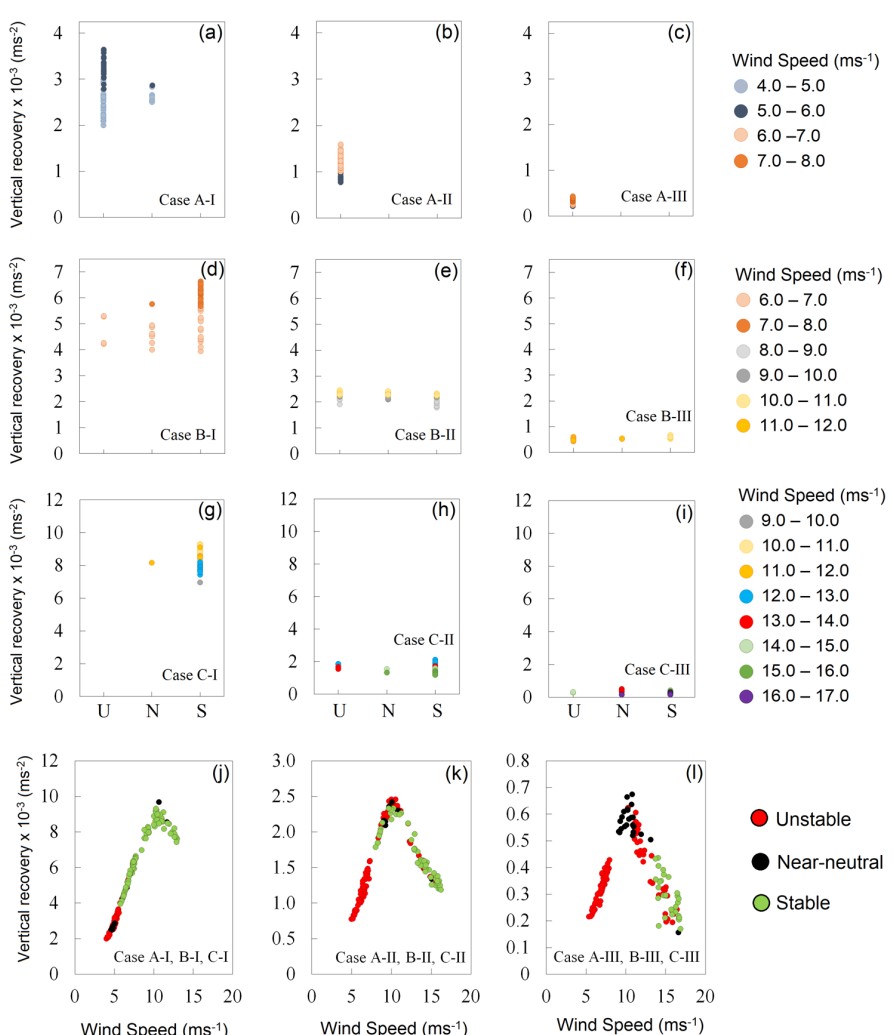

**Figure 10: Same as Fig. 9 but the stability classification is done using non-local lapse rate method.**

Fig. 10 also depicts the relationship of vertical recovery with stability and wind speed just like Fig. 9. However, in this case we estimated stability using the non-local lapse rate method instead of $R_f$. The results are quite similar to Fig. 9 showing that vertical recovery has a strong relationship with wind speed (Fig. 10j–l) but not with stability (Fig. 10a–i). These two figures, Fig. 9 and 10, clearly demonstrate that vertical recovery is dominated by wind speed but not with stability in our experiments.

**3.9 Effect of TKE advection on recovery processes**

Results show that the effect of horizontal advection of TKE on recovery is small. As expected, turning off horizontal TKE advection leads to changes in the vertical profile of TKE because TKE transport occurs only through vertical subgrid-scale turbulence. This leads to statistically significant changes in vertical recovery but the magnitude of the changes are small, up to 5% (Table 5). There is no significant effect on horizontal recovery. The spatial pattern of recovery (Fig. 11a) is similar to the experiments where horizontal TKE advection is tuned 'on'. Horizontal recovery is stronger at the upwind edges while vertical recovery is stronger in the interiors. However, the banded structure of horizontal recovery (Fig. 11b) is more prominent for TKE advection 'off' simulations.

**Table 5: Change (TKE advection 'off' – TKE advection 'on') in vertical recovery (x $10^{-3}$), $ms^{-2}$ and horizontal recovery (x $10^{-3}$), $ms^{-2}$, averaged over the 50 km x 50 km wind farm, the 48-hr simulation period and case A, B & C. The numbers in the parenthesis give the percentage change in recovery with respect to the corresponding momentum loss rate. * denotes that the numbers are significant at p<0.01**

| Case | I | II | III |
|---|---|---|---|
| $\Delta$Vertical Recovery ($ms^{-2}$) | 0.062* (-1.46%)* | 0.049* (-0.73%)* | 0.004* (-4.90%)* |
| $\Delta$Horizontal Recovery ($ms^{-2}$) | -0.022 (-0.5%) | -0.039 (-2.0%) | 0.02 (1.20%) |

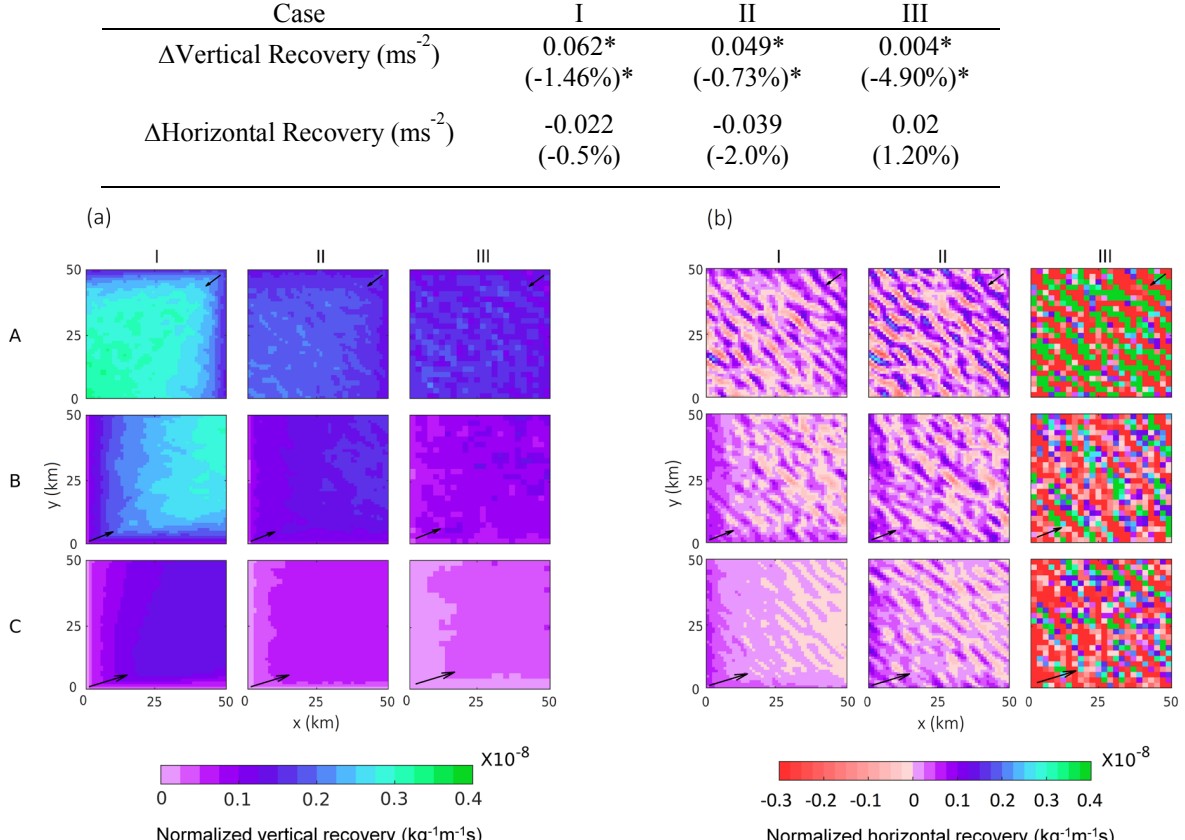

Figure 11: Same as Fig. 6 but with horizontal TKE advection turned 'off'.

## 4. Conclusions and Discussions

Turbines in a wind farm extract momentum from the flow and converts that to electricity. This study quantitatively explores the recovery processes through which the lost momentum is replenished so that wind farm can continue to function. For this purpose, a mesoscale model is used to simulate recovery processes in a hypothetical offshore wind farm. The model is equipped with a wind turbine parameterization and is driven by realistic initial and boundary conditions numerical experiments. The experiments quantify the recovery processes under different wind speeds and wind turbine spacings. The main conclusions of the study are as follows:

- Power generation in wind farms increases with increasing wind speeds with maxima at the upwind edges. The efficiency of wind farms increases with increasing inter-turbine spacings due to reduction in wake effects. In some cases with strong wind speeds, a wind farm with a 2 km spacing can operate at 100% efficiency because the low wake effects are not able to reduce the wind speeds below the rated speed.

- Vertical recovery is the main contributor to momentum replenishment in wind farms. Vertical transport of momentum by turbulent eddies into the turbine hub-height levels from above and below replenishes 70.8%–63.6% of the momentum loss in all but one cases. Momentum transport by horizontal advection plays a relatively weaker role replenishing only 6.6%–16.5% of the lost momentum. The relative contributions of horizontal and vertical recovery process are comparable only in the case with strong wind speeds and high inter-turbine spacing.

- The spatial patterns of vertical and horizontal recovery are complimentary. In general, vertical recovery is stronger in the interiors of the wind farms while horizontal recovery is stronger at the upwind edges. However, in sparsely spaced wind farms, horizontal recovery can also occur inside the wind farms as alternating bands normal to the direction of the flow.

- Vertical recovery increases with increasing wind farm density. It increases with increasing wind speed until the rated wind speed for the turbine after which, it starts to decrease. These systematic dependencies can be quantified using low-order empirical equations.

- Offshore wind farms do not alter the synoptic scale momentum fluxes in the atmosphere but can significantly alter mesoscale flow. Under strong wind conditions, a densely packed wind farm can generate a wake that is an order of magnitude longer than the size of the wind farm. Mesoscale circulations created by the wind farms can transport momentum downward into the ABL from aloft. Thus, they aid in recovery by making more momentum available for replenishment.

- Horizontal TKE advection has small effects on recovery processes in wind farms. Deactivating horizontal TKE advection in WRF leads to small (up to 5%) changes in vertical recovery but horizontal recovery and spatial patterns of recovery processes are not affected.

There are scopes for improving upon this study. First, the MYNN closure scheme in WRF does not account for horizontal turbulent transport thereby limiting our ability to fully quantify horizontal replenishment. This is a difficult problem to resolve because it will require the development of an advanced closure scheme. Second, the wind turbine parameterization calculates the TKE generated by wind turbines as the difference between the momentum extracted by the turbine and the power generated

by the turbine assuming that the electro-mechanical losses are negligible (Fitch et al., 2012). Archer et al. (2020) argues that electro-mechanical losses are not negligible. Comparing WRF and LES simulations, they contend that only about 25% of the difference between the momentum extraction and power generation goes to TKE while the rest is due to electro-mechanical losses. Hence, they suggested a correction factor of 0.25 for the coefficient that relates to turbulent kinetic energy ($C_{TKE}$). This issue needs in-depth investigation using observations to determine an appropriate partitioning between TKE generation and

electro-mechanical losses. If the electro-mechanical losses are as large as 3 times the TKE generated as suggested by Archer et al. (2020), those losses must be accounted for by adding a source term in the temperature equation in the model to ensure energy conservation. Third, in this study we investigated the role of horizontal TKE advection in recovery processes. However, to fully understand the importance of horizontal TKE advection, we must explore its effects on power production and wakes. Fourth, due to the relatively coarse spatial resolution of the model, subgrid-scale wake interactions are not represented.

Numerical experiments where the wake is explicitly parameterized (Volker et al., 2015) and wind farm layout effects are accounted for (Akbar and Porté-Agel, 2015) can improve our understanding of recovery processes. Finally, in this study we attempted to study the role of stability in recovery. However, for the limited set of data points in our simulations, we were unable to find a statistically significant dependence of recovery on stability as strong as its dependence on wind speed and inter-turbine spacings. More experiments need to be done with different stability cases and wind speed regimes to identify its

impact on recovery. However, wind speed and stability are often connected in the real world. For example, strong horizontal wind speeds rarely occur under unstable conditions (Stull, 2012; Wharton and Lundquist, 2012). Hence, we need to very careful about designing experiments with realistic boundary conditions to segregate the effect of stability and wind speed on recovery.

To the best of our knowledge, this is one of the first studies to look at wind farm replenishment processes under realistic

conditions including the role of mesoscale processes. The empirical equations for vertical recovery developed in this study can perhaps be used to develop parameterizations for replenishment in analytical wake models. Overall, this study is likely to significantly advance our understanding of recovery processes in wind farms and wind farm-ABL interactions.

**Code availability.** The numerical experiments were conducted with WRF that is a well-known open-source software available

in the public domain (https://github.com/wrf-model/WRF/releases/tag/v4.2.1). Model configuration files and changes required in the source code to calculate the recoveries are available at https://doi.org/10.5281/zenodo.4698913 (Gupta and Baidya Roy, 2021).

**Data availability.** The NCEP data used as initial and boundary conditions for the WRF model are available in the public domain (https://doi.org/10.5065/D6M043C6).

**Author contributions.** SBR conceptualized the study. TG and SBR formulated the experiment design. TG conducted the
numerical experiments and analyzed the results. TG and SBR wrote the manuscript.

**Competing Interests.** The authors declare that they have no conflict of interest.

**Acknowledgement.** The authors thank the IIT Delhi High-Performance Computing facility for providing computational
resources. The authors also sincerely thank the reviewers and editor for their valuable comments concerning the study.

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
