# Peer review of "Recovery Processes in a Large Offshore Wind Farm"

_Wind Energy Science, 2021_

## Author Comment (AC2)

**Response to Reviewer 1 – WES-2021-7**
**April 15, 2021**

We are thankful to the reviewer for providing the insightful feedback. Please find our response to the comments in blue in the section below.

The study presents numerical simulation runs by the WRF model to investigate recovery processes from an hypothetical 50x50 km$^2$ offshore wind farm. The WRF is driven by real weather data. The experiments quantify the recovery processes under different wind speeds and wind turbine spacing.

The study is well structured and written, although the language can be improved at some parts. The results are presented clearly, but some figures need be improved to be more accessible to the reader. The discussion picks up on the novel results but also remains very superficial at some points. My main criticism of the study is the neglection of the stability, which is a main control parameter for wake recovery. This is further explained in the comment sections. With the consideration of this aspect and further improvement, the study can be a very valuable contribution to the offshore wind energy community.

**General comments**

The abstract gives a detailed description of the study, but remains very vague on the results. The results need to be presented more precisely and concretely, (e.g how is high inter turbine spacing defined, densely packed etc…).  For more details please see the specific comments.

Thank you for pointing this out. We will make the abstract more clear in terms of results by providing more quantitative information.

My main criticism is that the study does not take into account the stability and boundary-layer height regarding wake recovery. For the description of offshore wind farm impacts on the atmosphere (such as far wake effects/blockage effect or the influence of the farm on vertical turbulent moment flux, as mentioned on p.13 l.24), a consideration of the stability and ABL top (as mentioned in p.10 l.25) as major parameter, along with the park layout and wind speed, have been identified in several recent studies (e.g. Djath et al . 2018, Siedesleben et al . 2018, Cañadillas et al. 2019, Platis et al. 2020 etc…). However, this study only takes turbine spacing and the wind speed into account.

In addition, the results from these mentioned studies and further recent studies about the ongoing investigation of the far field effects of offshore wind farms are not addressed in the study. Also on p.13 l.24: Turbulent vertical mixing depends on the thermal and dynamic stability.  Also for strong horizontal wind speed during strong convective conditions vertical recovery may remain the main

contributor. Therefore, I highly suggest at least to take the stability (e.g. lapse rate or Richardson bulk number) for the investigated cases into account and include them in the discussion.

This is a very important point. While designing the experiments to study recovery, we wanted to look at wind farm design, specifically, the variability of recovery process under different wind farm spacing and wind speeds. But we agree with the reviewer that the stability question is extremely important. As suggested by the reviewer, we will take an extensive look at the static and dynamic stability patterns of the 3 case studies using lapse rate, Richardson number and other appropriate metrics.

Figures have to be re-worked including units to the scales. For some figures I also recommend to enlarge them to make them more readable. Also the range of the scale hat to be rearranged as e.g in figure 6 a) the variation is seen hardly.

We understand the reviewer's concern regarding the figures. We will replot the figures to make them easier to understand. We will add units to the scales in Fig. 3, 4 and 6. If we enlarge the figures, we will not be able to fit them in the same panel, making it difficult to compare between cases. Hence, we will replot Fig. 3c and Fig. 6 with RAINBOW type colormap to enhance visual clarity. We will also enlarge the text in the figures to make them legible.

To make things easier for the reader, I suggest to define the mean wind direction as positive x. This will help the reader to compare easier cases A with B and C. This will also help to better distinguished between flow effect parallel and perpendicular to the flow e.g., the interpretation of Figures 3 b) , 4, 5 and 6.

We understand that this could be confusing for the readers. But we prefer to keep the figures unchanged because we want to give the readers a flavor of the fact that these simulations are constrained by observed boundary conditions where the wind directions vary. We have taken the components of meteorological variables along the dominant direction. To help readers understand these figures better, we have added arrows showing the prominent wind directions. We will make these arrows bolder so that the difference in wind direction between the different cases clearly discernible.

Why is the recovery only presented for the wind farm domain (Fig. 6). I expect also far field effects similar to the wake effects to be seen in the vicinity of the wind farm. Also this will give a broader picture of the upwind and downwind effects.

Our primary purpose in this study is to show the recovery in the wind farms. The study was conceptualized with the aim to study how wind speed recovery happens in the wind farm. It is important to study recovery in wind farms because it also has a practical implication. It helps us understand how replenishment of energy allows a spatially large wind farm to function. Far-field effects are no doubt very interesting but beyond the scope of the current study.

**Specific comments**

p.1 l.14. How is high defined? Narrow spacing? Please be more precise.

In our experiments, inter-turbine spacings range from 0.5 km (densely-packed Case I) to 2 km (sparsely-packed, Case III). We will add this quantitative information on spacing to the abstract.

p.1 l.16: What is meant by can be quantified using low-order empirical equations? Please be more concrete.

We have quantified the vertical recovery using second-order empirical equations. The details are given in section 3.5.2 of the paper. We will modify the statement to make it more concrete.

p. l.17. What is meant by high wind speed. Which range are you referring to?

In this study, apart from the different inter-turbine spacings, we also explored the role of different wind speed ranges over which the wind turbines operate, ranging from lowest wind speed of 3 ms$^{-1}$ to highest wind speed of 18 ms$^{-1}$. Case A corresponds to low wind speed range and case C represents the high wind speed range. We will add this quantitative information on wind speed to the abstract.

p.2. l.15: What version of the WRF model? Which wind turbine parameterization ? Please rephrase the sentence in the abstract or in the introduction as they are identical.

We used WRF version 4.2.1 and the Fitch et al. (2012) wind farm parameterization for our simulations. We will add the WRF version and turbine parameterization in the paper. We will also rephrase the sentences in the introduction to avoid any duplication.

p.4. l14ff: I suggest to add a figure showing the relation between grid cell and turbine spacing in order to makes things more clear for the reader

We will add a figure showing the locations of turbines within the grid cells to make the relation between grid cell and turbine spacing clearer to the readers.

p. 5 l.15: This is still a simulation, so the term 'realistic' is not appropriate.

We will remove the term 'realistic' from the mentioned sentence.

p.6. l.5: Please introduce here what k,i,j is referring to.

The letters k, i, and j are the location indices in the vertical, zonal and meridional directions. We will explain this in the text.

p.6. l.18:Over which domain are the horizontally averaged? Over the wind farm domain?

The horizontal averaging is over whole of domain 3 from the WRF simulations. We will explain this in the manuscript.

Eq. 8 Please describe what is defined by î and Äµ ?

The symbols î and j are the unit vectors in the zonal and meridional directions, respectively. We will explain this in the manuscript. We are unable understand the second symbol. We assume that it is a typo and the reviewer means j.

p.9. l15: Please give a broader description about the statistical analysis.

We will add a description and reference about the Wilcoxon sign rank test.

Eq. 15: The denominator on the right-hand side of Eq. 9 should contain the unit. I also suggest to write 140m – 28m

Adding units to only one term in the equation will not be appropriate. We think the way we have written the equation with m in the numerator is causing some confusion. We will rewrite this equation using symbols to eliminate any confusion about units.

p.7 l.25 Why small v?

Sorry about the typo. We will correct it in the manuscript.

Fig. 3. The figure is hard to read, especially 3c). Please enlarge the plots and the labels. Please add a unit to the scales. For case A the resolution is way to small to be able to follow the analysis on page. 9. l. 15-16.

As mentioned earlier, enlarging the figures will make it impossible to fit them in the same panel making it hard to compare between cases. We will replot the figure with a RAINBOW type colour scheme for better visibility and add units to the labels.

p.10. l.5: Not true several other studies such as Platis et al. 2018, 2020, Siedersleben et al. 2018 reported a deceleration of up to 40 % in the wake of offshore wind farms.

The values reported in our manuscript are averages. The studies mentioned by the reviewers report maximum deceleration, e.g., Platis et al., 2018, 2020 found a maximum deceleration of 40% and 43% in the wake of offshore wind farms, respectively. However, we could not find any explicit mention of deceleration rates in Siedersleben et al., 2018.  We will add this comparison in the text.

p.10 l.25. What is the height of the ABL top?

The ABL top is ~1400 m for Case A and ~1000 m for Case B & C. We will add a line showing the ABL top height in Fig. 4.

Fig. 4. I recommend to mark the area where the wind farm is located.

The wind farm is already marked with a black dashed box in Fig. 4. It is also mentioned in the figure caption. We will make the dashed box that depict the wind farm cross-section bolder for better visibility.

Fig. 4: Case A III. Why is there a deceleration and then an acceleration of the flow between 0-1000 m and at x= 780-850 km?

Please note that the direction of wind flow is from right to left in Case AIII. Hence, the regions with red color between 0-1000 m and at x= 780-850 km indicate deceleration upwind of the farm as seen in all cases. In this particular case, there is an alternating band of white and red. This pattern is perhaps giving the impression that there are alternating bands of acceleration and deceleration. However, the white colored regions do not indicate acceleration. Rather, they denote regions where the signals are not statistically significant. Thus, this pattern actually indicates that the upwind deceleration in case AIII is relatively weak. We will explain this in the text and also add an explanation of the white patches in the figure caption.

Fig. 4 The upwind deceleration seem very impressive. I am wondering whether a too small simulation domain is causing an intensification by boundary reflections? Is there a way to assess this influence ?

The domain is very large, 1500 km X 1500 km with a 50 km X 50 km wind farm in centre. The acceleration is not caused by boundary reflections. We had conducted sensitivity studies with domains of different sizes starting from a 300 km X 300 km domain. We found some wake reflections in the smaller domains but there were no boundary effects with this large domain. We will add this explanation in Section 2.2 of the paper.

Fig. 4 b. Why is a streak pattern visible? Can this be also attributed to artifacts caused by the simulation?

Streak patterns are updrafts and downdrafts induced due to wind farm. We conducted significance tests on the results and plotted only the statistically significant signal to minimize the depiction of random noise.

p.11.9 ff. I do not understand how this argument contributes to case C-I. Please describe more clearly.

We assume the reviewer is talking about p11 L8 because P11 L9 talks about case III. We think this sentence is superfluous. We will remove this so as to no cause confusion.

Fig. 5 I do not understand the meaning of the legend at the second left figure in the first row.

These refer to the synoptic, mesoscale and microscale fluxes calculated using the equations described in Section 2.3.1. We will also clarify this in the figure caption.

Fig .5. The description and argumentation of the results is at some points not very precise.

We will expand Section 3.4 to provide improve our explanation of the pattern of vertical synoptic, mesoscale and microscale fluxes in a wind farm.

p.11. l.17. It would be helpful for the reader just to mention again very briefly the difference between synoptic and micro scale.

We will add a brief description of the scales and refer to the appropriate equations in the text.

p.11 l.8-10. "It is possible that this meso-scale momentum transport aids in the wind farm recovery by making more momentum available for downward mixing by turbulence." This is very speculative. Is there a way to justify it ? Can the variations at different heights be explained?

We agree with the reviewer that this statement appears speculative. Indeed, that is why we have intentionally used the word 'possible' because the evidence for this statement is limited to only case C-I. In Case C-I, the UW mesoscale flux is maximum at ~1500 m altitude showing that the momentum is transferred from above free atmosphere (above the PBL height of ~1000 m) into the boundary layer. In Case C-I, a negative UW mesoscale flux at 1500 m height depict a net downward transport of higher momentum. The mesoscale flux is negligible up to 1000 m (around PBL height in the wind farm) because within the wind farm no vertical mesoscale momentum transport leads to downward transport of momentum rather the downward transport of momentum happens through microscale fluxes. We will expand this entire section to further clarify our point. Moreover, we will also provide detailed explanation of the microscale flux patterns.

Fig. 6: Why does the plots in case of III look much coarser than for I and II ?

The reviewer is absolutely right. In case III, we have a turbine in every other grid cell to ensure a 2 km spacing. The data points in in this case are averaged over 4 grid cells covering 2 km X 2 km. That is why case III looks smoother. We will add this explanation to the text.

Fig. 7a) Why is there such a sharp boundary (jump) at about x= 4x10-3 ? Because of the different cases? This could be mentioned in the text.

Yes, the jump at x= $4x10^{-3}$ is for the different cases. We will mention this in text.

p. 17 l. 11. Please refer to the equation or describe the integration by an separate equation.

We noticed there is a mistake in the text, it should have been section 2.3.1 instead of 2.3.3. As suggested by the reviewer, we will add the equation number for the reader's reference.

p.18 l. 31. What do you mean exactly by synoptic scale effects?

What we meant to say here is that wind farms do not affect any synoptic scale fluxes as evident from section 3.4 Fig. 5. In the mentioned figure, the effect on synoptic scale fluxes by the wind farm is negligible. This is the reason the synoptic scale flux line is not visible in the figure. We will state this more clearly in this bullet point.

**Technical corrections**

p.2. l.1 ff: Please put the citations in chronological order: Will be done.

p.4 l.9: Please correct: boundary-layer scheme: Will be corrected.

p.4 l.10: Please correct: second-order moments: Will be corrected.

p.6 l.22 typo error 'that': Will be corrected.

Fig. 7 b)-d) Please title the figures with its specific cases. : The specific cases will be added in the figure as titles.

References:

Djath, B., Schulz-Stellenfleth, J., and Cañadillas, B.: Impact of atmospheric stability on X-band and C-band synthetic aperture radar imageryof offshore windpark wakes, Journal of Renewable and Sustainable Energy, 10, 043 301, 2018

Cañadillas, B., Foreman, R., Barth, V., Platis, A., Siedersleben, S. K., Bange, J., Lampert, A., Bärfuss, K., Hankers, R., Schulz-Stellenfleth, J.,Djath, B., Emeis, S., and Neumann, T.: Offshore wind farm wake recovery: Airborne measurements and its representation in engineeringmodels, Wind Energy, https://doi.org/10.1002/we.2484, 2019.

Platis, A., Hundhausen, M., Siedersleben, S. K., Lampert, A., Bärfuss, K., Schulz-Stellenfleth, J., Djath, B., Emeis, S., Neumann, T., Cañadil-385las, B., and Bange, J.: Long-range modifications of the wind field by offshore wind parks - results of the project WIPAFF, MetZet, 2020b

Siedersleben, S. K., Lundquist, J. K., Platis, A., Bange, J., Bärfuss, K., Lampert, A., Cañadillas, B., Neumann, T., and Emeis, S.: Micrometeorological impacts of offshore wind farms as seen in observations and simulations, Environmental Research Letters, 2018

**References**

Fitch, A. C., Olson, J. B., Lundquist, J. K., Dudhia, J., Gupta, A. K., Michalakes, J., and Barstad, I.: Local and mesoscale impacts of wind farms as parameterized in a mesoscale NWP model, Mon. Weather Rev., 140, 3017–3038, https://doi.org/10.1175/MWR-D-11-00352.1, 2012.

---

## Author Comment (AC3)

**Response to Reviewer 2 – WES-2021-7**
**April 15, 2021**

We are thankful to the reviewer for providing the insightful feedback. Please find our response to the comments in blue in the section below.

This study explores different recovery processes in a large hypothetical offshore wind farm in the Arabian Sea for 3 different meteorological conditions using the mesoscale model WRF and the Fitch WFP. The manuscript is well written in general. Some figures could be improved to faciliate the undestanding of the reader. I think it is an interesting study that should potentially be published in WES, after carefully addressing the comments below.

**General comments**

My main criticism of this study is that, while the three different cases certainly consider different wind speed ranges, they might also have different stability ranges. However, the stability for the cases is not discussed (see specific comments below), although various studies have shown an impact of stability on wakes (Lee and Lundquist 2017, Cañadillas et al. (2020) to mention just a few).

This is a very important point. While designing the experiments to study recovery, we wanted to look at wind farm design, specifically, the variability of recovery process under different wind farm spacing and wind speeds. But we agree with the reviewer that the stability question is extremely important. As suggested by the reviewer, we will take an extensive look at the static and dynamic stability patterns of the 3 case studies using lapse rate, Richardson number and other appropriate metrics.

My second concern is the sensitivity of the model results with respect to the vertical resolution. The vertical resolution is rather coarse (>20 m) compared to the resolution, which is suggested to be necessary to capture wind farm effects (e.g. Siedersleben et al 2020, Tomaszewski and Lundquist (2020) or Pryor et al. (2020)). In addition Pryor et al. (2020) also pointed out that the TKE magnitude depends on the vertical resolution. One of your conclusions concerns the effect of TKE advection on the recovery process. This conclusions could be faulty, if the resolution is too coarse. Please elaborate.

Thank you for bringing these studies to our notice.

Tomaszewski and Lundquist (2020) suggested a low-level vertical resolution of ~10 m and horizontal of either 1 km or 3 km in order to achieve the appropriate mixing required to match the expected surface warming and drying. Siedersleben et al. (2020) clearly recommended a vertical level on the order of ~12m. However, they also mentioned that in case of limited computational resources, a horizontal resolution of 5 km and vertical resolution of 35 m below 100 m also captures the most important features

over the wind farm. Also, Pryor et al. (2020) showed an impact of vertical resolution on TKE added in the wind farm and its vertical distribution.

Our simulations are computationally very expensive with each simulation costing ~16000 computational hours on a high-performance computer with 200 processors. We have conducted 24 simulations leading to a total of 384000 computing hours. Each simulation requires extensively large amount of computation resources because we have a very large 1500 km X 1500 km horizontal domain discretised with a high horizontal grid spacing (1 km for the finest grid). This configuration is essential for our simulations. We need the large horizontal domain to avoid wake boundary reflections and the fine horizontal resolution to capture the intra-wind farm wake effects for different inter-turbine spacings. We have used a vertical resolution of ~21.4 m in the lowest 150 m of model domain. This resolution is comparable to Pryor et al., 2020 (~16.7 m) and is  appropriate for capturing the important features including TKE over wind farms in case of limited computational resources (Siedersleben et al., 2020).

Lastly, we have used total 61 vertical levels till top of the atmosphere. It is necessary to keep a fine resolution beyond the rotor depth (lowest 150 m) as well because it will help us capture the effect of windfarm on mesoscale and synoptic fluxes also.

**Specific comments**

p 1, line 27ff: Is the citation of so many studies necessary here?

We have tried to provide an exhaustive list of studies that have explicitly discussed how recovery happens in the wind farms through microscale and mesoscale processes. Some of these studies have only mentioned the issue in passing while others have explored it using rigorous quantitative approaches. We will reduce the number of citations to more quantitative studies.

P 2, line 8: "… also confirmed by LES…", in line 5 you write that the recovery process was previously investigated by LES simulations. Why is the study by Calaf et al. (2010) listed separately from the others?

Sorry for missing the reference of Calaf et al. (2010) in P2L6 with other citations. We will add the mentioned reference with others in P2L6.

P 2, line 27: Here you write that TKE advection is deactivated in your study, while later on Page 5, line 14-17 you write that both activated and deactivated scenarios are used. Please make these statements consistent.

We agree that this sentence should be rephrased for clarity. We conducted two sets of simulations: the first set with TKE advection on and the second with TKE advection off. All results shown in sections 3.1

to 3.6 include TKE advection. In order to study the effect of TKE advection like Archer et al. (2020), we conducted sensitivity simulations with the TKE advection turned off. The comparison of these two sets of simulations is discussed in section 3.7. We will make this clearer in the revised manuscript.

P 3, line 3: Please provide the version number of the WRF version that you have used. The bug found by Archer et al. (2020) showed, how important it is to document clearly, which version number has been used.

We have used WRF Vr 4.2.1 for our simulations. We will add this in the updated manuscript.

P3, line 19: How transferable are your results to other regions? This farm is quite far offshore. Existing offshore farms are closer to the coast. How could coastal effects change your results?

The results are valid for all deep offshore regions if the basic meteorology is same. However, in case of some extreme phenomena such as cyclones, we anticipate that results may vary. Coastal wind farms are often affected by sea breezes and can have different recovery patterns. We will mention this is the discussion section. We are currently conducting a study for coastal wind farms but we do not have final results yet.

P3, line 24: A vertical resolution of > 20 m is at the upper limit of the necessary resolution to capture wind farm effects correctly as studies by Siedersleben et al 2020, Tomaszewski and Lundquist (2020) or Pryor et al. (2020) have indicated. In addition Pryor et al. (2020) also pointed out that the TKE magnitude depends on the vertical resolution. How sensitive are your results to vertical resolution?

Please see the justification given above in the general comments section.

P4, line 4: Was the Sea Surface Temperature also taken from that source?

Yes, the SST was also taken from the National Centers for Environmental Prediction Final Operational Global Analyses dataset (NCEP, 2015). We will mention this in the revised manuscript.

P4, line 4-11: A tabular overview could provide a better overview on the set-up

We will provide the physics parameterization that are used in the simulations in a tabular form in revised manuscript.

P4, line 14: This is a rather small turbine compared to turbines that are currently being installed offshore. Why have you used such a small turbine and how representative are your results for more modern turbines?

We understand the reviewer's concern. The absolute magnitude of the power generation and recovery will depend on the turbine rating. However, we expect that the spatial patterns and relative magnitudes of the recovery will be valid for any of the installed turbine size provided that the inter-turbine distances

and meteorological conditions are comparable. Please note that we have presented the recovery terms as a function of power generated in Figure 6 and Table 2.

P5, line 7: Atmospheric stability is another relevant parameter for wind energy. Is the stability similar in all three cases? Otherwise this will affect the conclusions as well.

We understand the reviewer's concern. We will analyse the stability conditions for all the three cases and add a sub-section in the revised manuscript.

P 6, Eq 6: Define what hat^i and hat^j mean. Consider to add subindex "h" to indicate that hub-height winds are used here.

The symbols î and ĵ are the unit vectors in the zonal and meridional directions, respectively. We will explain this in the manuscript. We will add a sub-index "h" to indicate the hub-height wind speeds in order to avoid any confusions with Eq. (4) & (5) for the readers.

P 8, line 18: How do these values compare with Volker et al. (2017)

We cannot directly compare Volker et al. (2017) with our study because the wind farm size and number of wind turbines are different than ours. However, the central tendencies of wind speeds are somewhat comparable. Here we present some very approximate comparisons by visual interpolation:

1) In our case, we are getting 31% efficiency for 4.5D and 11.3 m/s mean CTRL case wind speed. In case of Volker et al. 47.7% (interpolated) efficiency for 5.25 D and at a wind speed of 9.1 m/s.
2) In our case, we are getting 81% efficiency for 4.5D and 15.6 m/s mean CTRL case wind speed. In case of Volker et al. 60.3% (interpolated) efficiency for 5.25 D and at a wind speed of 13.1 m/s.
3) In our case, we are getting 78% efficiency for 8.9D and 11.3 m/s mean CTRL case wind speed. In case of Volker et al. 77.5% (interpolated) efficiency for 10.5 D and at a wind speed of 9.1 m/s.
4) In our case, we are getting 100% efficiency for 8.9D and 15.6 m/s mean CTRL case wind speed. In case of Volker et al. 85% (interpolated) efficiency for 10.5 D and at a wind speed of 13.1 m/s.

These comparisons are too approximate to be presented in the paper.

Figure 3, 4, 6: Please add variables and units to the colorbar.

We will add variables and units to the colorbars in the mentioned figures.

Figure 3: Are averages over the entire two or three day period shown in that figure? Or is it a snapshot? Please clarify. Consider to add the "plots depict the wind farm only, that is the black square in (c), if appropriate". In figure (c), please add: every xth vector is shown

In Fig. 3 (b) and (c), averages over the entire two day period is shown. We will add the clarification in manuscript and modify the captions as per the reviewer's suggestion.

P 9, line 13: here it says averaged over the rotor depth, while in the caption of figure 3 "hub-height wind" is mentioned. Please clarify.

We apologize for the mistake. The plot shows differences in wind speeds averaged over the rotor depth, not the hub-height wind. We will correct the error in the Fig. 3 (c) caption.

P 9, line 17: The wake length depends also strongly on stability. Again, have you checked that the stability is the same in all simulations?

We will analyse the stability for the different cases and address the concern in revised manuscript.

P 10, line 3: Wake lengths can be defined in many ways. You mention a wake length of 521 km, which is very long. Please add your definition of a wake length.

We take the difference the hub-height wind speeds between the control and wind farm cases and estimate the statistical significance using the Wilcoxon sign rank test. The wake lengths are calculated as distance from the wind farm to the grid cell where the difference falls below the 99% level of significance . We will add this in the manuscript.

P 10, line 13 – 15 and Figure 4: The figure could be easier understandable, if Case A would be turned into the mean wind direction and thus the colours would be the same in all cases. Please consider doing so.

We understand that this could be confusing for the readers. But we prefer to keep the figures unchanged because we want to give the readers a flavor of the fact that these simulations are constrained by observed boundary conditions where the wind directions vary. We have taken the components of meteorological variables along the dominant direction. To help readers understand these figures better, we have added arrows showing the prominent wind directions. We will make these arrows bolder so that the difference in wind direction between the different cases clearly discernible.

Figure 7 (a): Most points follow the linear trend. However there are also quite some points with relatively high momentum loss rate but small vertical recovery (especially around 10*10⁻³ msⁿ²). During which situations does this behaviour occur? Also there is a sharp drop at about 3*10⁻³ msⁿ², where does this come from?

There are also some points where though the momentum loss rate is high, the vertical recovery is low. These are the points on the upwind edges of the wind farm where the horizontal recovery dominates leading to lower values of vertical recovery. The sharp change at momentum loss rate ~ $3.5 \times 10^{-3}$ in Fig. 7 (a), is because of change in cases. We will explain this in the revised manuscript.

Figure 8: Why is only the U-momentum flux considered here, even though the wind direction is south-west / north-east?

Figure 8 presented here is for Case C-I. This is the only case where the mesoscale fluxes are of the same order of magnitude as turbulent fluxes. The wind direction for case C is west-south-westerly with stronger u components than v. This is the reason the u momentum flux is much higher than the v momentum flux (P13L12) and hence, only the u-momentum flux is considered in the paper. We will add this explanation in the section 3.7.

Section 3.7: I think this section deserves a bit more elaboration and possibly a figure. As you mention in your introduction on page 2, there has been a debate, whether TKE advection needs to be activated or not. With that in mind, it would be good to give more evidence for your conclusion on TKE advection.

We will add a table in the section 3.7 with values of difference in horizontal and recovery for the different cases analysed. We will also add a figure showing spatial patterns of vertical and horizontal recovery with TKE advection off.

**Technical corrections**

P5, line 21: Consider to add "the method by" in using Avissar and Chen (1993).

We will add "the method by" in P5 L21.

P6, line 22: "that" is used before and after the citation.

Sorry for the mistake, we will correct it.

P8, line 7: "As" should not be capitalized.

Sorry for typo, we will correct it.

P8, line 23-24: Fig. 3b is referenced twice within one sentence

We will correct the reference of Fig. 3b in P8L23-34.

Figure 3: sub-figures are usually referenced only with (b) not with Fig. 3 (b). Please check with the journal guidelines

We thank the reviewer for pointing this out. Though we cannot find specific instruction in the journal guidelines, we checked the papers published in the journal and they follow the same style of figure referencing in the caption as mentioned by the reviewer. Therefore, we will revise the figure captions as per the reviewer's suggestion.

P 13, line 22: consider to add "side" or the like after "at the upwind"

We will add the word "side" after upwind.

P 18, line 28: "till" is rather colloquial. Consider to use "until" instead of "till"

We will replace till by until in P18L28.

P 19, line 24 – 28: I agree with the authors that it is an interesting study, but how significant a contribution is cannot be known before publishing. I would suggest to advertise it a bit less.

We will rewrite this paragraph as per the reviewer's suggestion.

P 19, line 31: Code availability: Please add the URL to the WRF github repository (or where ever you downloaded WRF)

We had downloaded WRF version 4.2.1 from github. We will add a URL (https://github.com/wrf-model/WRF/releases/tag/v4.2.1) for the same.

P 19, line 32: Code availability: Please also add namelist.wps to the the repository. Why are only the example files for 0.5 km included in the repository, even if they are modified within the code it would be good to have them all. Please also add a description of model changes in the 5 files that you mention in the README.

We will add all the files suggested by the reviewer to the repository. We will list all changes made in the model code in the README file.

P 20, line 1-2: Please add a link to the NCEP data.

We will add a link to NCEP data in the revised manuscript.

P 21, line 10: Chou et al: How can this publication be accessed? Please provide a doi / url

We will add a url (https://ntrs.nasa.gov/citations/20010072848) for Chou et al. in the revised manuscript.

P 21, line 30-31: Hong et al: How can this publication be accessed? Please provide a doi / url

We will add the url
(https://www.kci.go.kr/kciportal/ci/sereArticleSearch/ciSereArtiView.kci?sereArticleSearchBean.artiId=ART001017491) for Hong et al. in the revised manuscript.

P 22, line 22: line 1-2: IRENA: How can this publication be accessed? Please add a url.

We will add a url  (https://www.irena.org/publications/2019/Oct/Future-of-wind) for the IRENA technical report.

P. 22, line 7: Li et al: How can this publication be accessed? Please provide a doi / url

We will add the url (https://doi.org/10.1007/s00376-010-0041-0) for this publication in the paper.

P. 22, line 16: Noppel et al: How can this publication be accessed? Please provide a doi / url

We will add the url (https://doi.org/10.1023/A:1015556228119) for this publication in the paper.

P 22, line 26: Skamarock et al: How can this publication be accessed? Please provide a doi / url. In addition: your zenodo repository indicates that you have been using WRF 4.2.1, why do you cite version 3 here?

We apologize for this mistake. The correct citation should be of: Skamarock, W. C., Klemp, J. B., Dudhia, J., Gill, D. O., Liu, Z., Berner, J., … Huang, X. -yu. (2019). A Description of the Advanced Research WRF Model Version 4 (No. NCAR/TN-556+STR). doi:10.5065/1dfh-6p97. We will correct this in the manuscript.

P. 23 line 5: Zeng et al: How can this publication be accessed? Please provide a doi / url

We will add the url (https://doi.org/10.1175/1520-442(1995)008<1156:LIAFAI>2.0.CO;2) for this publication in the paper.

**References**

Cañadillas B, Foreman R, Barth V, Siedersleben S, Lampert A, Platis A, Djath B, Schulz-Stellenfleth J, Bange J, Emeis S, Neumann T (2020) Offshore wind farm wake recovery: Airborne measurements and its representation in engineering models. Wind Energy 23(5):1249–1265, DOI 10.1002/we.2484

Lee JCY, Lundquist JK (2017b) Observing and Simulating Wind-Turbine Wakes During the Evening Transition. Boundary-Layer Meteorology 164(3):449–474, DOI 10.1007/s10546-017-0257-y

Siedersleben, S.K., Platis, A., Lundquist, J.K., Djath, B., Lampert, A., Bärfuss, K., Cañadillas, B., Schulz-Stellenfleth, J., Bange, J., Neumann, T. and Emeis, S.: Turbulent kinetic energy over large

offshore wind farms observed and simulated by the mesoscale model WRF (3.8.1), Geosci. Model Dev. (Online), 13(NREL-JA-5000-75874), https://doi.org/10.5194/gmd-13-249-2020, 2020.

Tomaszewski JM, Lundquist JK (2019) Simulated wind farm wake sensitivity to configuration choices in the Weather Research and Forecasting model version 3.8.1. Geoscientific Model Development Discussions, Final version published after completion of this study at DOI 105194/gmd-13-2645-2020 DOI 10.5194/gmd-2019-302

Pryor SC, Shepherd TJ, Volker PJH, Hahmann AN, Barthelmie RJ (2020) "Wind Theft" from Onshore Wind Turbine Arrays: Sensitivity to Wind Farm Parameterization and Resolution. Journal of Applied Meteorology and Climatology 59(1):153–174, DOI 10.1175/jamc-d-19-0235.1

Volker PJH, Hahmann AN, Badger J, Jørgensen HE (2017) Prospects for generating electricity by large onshore and offshore wind farms. Environmental Research Letters 12(3), DOI 10.1088/1748-9326/aa5d86

---

## Author Response (AR1)

**Authors' Response to the Reviews**

We are thankful to the reviewers for providing the insightful feedback. Please find our response to the comments in blue in the section below. We have also highlighted the changes made in the manuscript and given the corresponding page and line numbers for reference.

**Reviewer 1 – WES-2021-7**

The study presents numerical simulation runs by the WRF model to investigate recovery processes from an hypothetical 50x50 $km^2$ offshore wind farm. The WRF is driven by real weather data. The experiments quantify the recovery processes under different wind speeds and wind turbine spacing.

The study is well structured and written, although the language can be improved at some parts. The results are presented clearly, but some figures need be improved to be more accessible to the reader. The discussion picks up on the novel results but also remains very superficial at some points. My main criticism of the study is the neglection of the stability, which is a main control parameter for wake recovery. This is further explained in the comment sections. With the consideration of this aspect and further improvement, the study can be a very valuable contribution to the offshore wind energy community.

**General comments**

The abstract gives a detailed description of the study, but remains very vague on the results. The results need to be presented more precisely and concretely, (e.g how is high inter turbine spacing defined, densely packed etc…).  For more details please see the specific comments.

Thank you for pointing this out. We have made the abstract clearer by providing more quantitative information in terms of spacing and wind speed range (P1, L13–16 and P1,L18).

My main criticism is that the study does not take into account the stability and boundary-layer height regarding wake recovery. For the description of offshore wind farm impacts on the atmosphere (such as far wake effects/blockage effect or the influence of the farm on vertical turbulent moment flux, as mentioned on p.13 l.24), a consideration of the stability and ABL top (as mentioned in p.10 l.25) as major parameter, along with the park layout and wind speed, have been identified in several recent studies (e.g. Djath et al . 2018, Siedesleben et al . 2018, Cañadillas et al. 2019, Platis et al. 2020 etc…). However, this study only takes turbine spacing and the wind speed into account.

In addition, the results from these mentioned studies and further recent studies about the ongoing investigation of the far field effects of offshore wind farms are not addressed in the study. Also on p.13 l.24: Turbulent vertical mixing depends on the thermal and dynamic stability.  Also for strong horizontal wind speed during strong convective conditions vertical recovery may remain the main

contributor. Therefore, I highly suggest at least to take the stability (e.g. lapse rate or Richardson bulk number) for the investigated cases into account and include them in the discussion.

This is a very important point. While designing the experiments to study recovery, we wanted to look at wind farm design, specifically, the variability of recovery process under different wind farm spacing and wind speeds. But we agree with the reviewer that the stability question is extremely important. As suggested by the reviewer, we have taken an extensive look at the static and dynamic stability patterns of the 3 case studies using lapse rate and Richardson number. We found that all three cases are dynamically unstable. Case A is predominantly statically unstable. Case B is a mix of statically unstable and stable atmosphere, and Case C predominantly statically stable. We have added a section 3.2 (P10, L1 –L14) in the revised manuscript describing the stability conditions. We have also added section 3.8 (P19, L4 –L21) showing effect of stability on vertical recovery. In the discussion section, we have addressed the challenges in doing such simulations under realistic boundary conditions (P23, L33 – P24, L6). We have also added the PBL height in Fig. 4.

Figures have to be re-worked including units to the scales. For some figures I also recommend to enlarge them to make them more readable. Also the range of the scale hat to be rearranged as e.g in figure 6 a) the variation is seen hardly.

We understand the reviewer's concern regarding the figures. We have re-plotted the Fig. 3c and 6 to make them easier to understand. We have further added units to the scales in Fig. 3, 4 and 6. If we enlarge the figures, we will not be able to fit them in the same panel, making it difficult to compare between cases. Hence, we have re-plotted Fig. 3c and Fig. 6 with RAINBOW type colormap to enhance visual clarity.

To make things easier for the reader, I suggest to define the mean wind direction as positive x. This will help the reader to compare easier cases A with B and C. This will also help to better distinguished between flow effect parallel and perpendicular to the flow e.g., the interpretation of Figures 3 b) , 4, 5 and 6.

We understand that this could be confusing for the readers. But we prefer to keep the figure axes unchanged because we want to give the readers a flavour of the fact that these simulations are constrained by observed boundary conditions where the wind directions vary. We have taken the components of meteorological variables along the dominant direction. To help readers understand these figures better, we had added arrows showing the prominent wind directions. We have now made these arrows bolder so that the difference in wind direction between the different cases is clearly discernible.

Why is the recovery only presented for the wind farm domain (Fig. 6). I expect also far field effects

similar to the wake effects to be seen in the vicinity of the wind farm. Also this will give a broader picture of the upwind and downwind effects.

Our primary purpose in this study is to show the recovery in the wind farms. The study was conceptualized with the aim to study how wind speed recovery happens in the wind farm. It is important to study recovery in wind farms because it also has a practical implication. It helps us understand how replenishment of energy allows a spatially large wind farm to function. Far-field effects are no doubt very interesting but beyond the scope of the current study.

**Specific comments**

p.1 l.14. How is high defined? Narrow spacing? Please be more precise.

In our experiments, inter-turbine spacings range from 0.5 km (low inter-turbine spacing/densely-packed: Case I) to 2 km (high inter-turbine spacing/sparsely-packed, Case III). We have added this quantitative information on spacing to the abstract (P1, L13–14).

p.1 l.16: What is meant by can be quantified using low-order empirical equations? Please be more concrete.

We have quantified the vertical recovery using second-order empirical equations. The details are given in section 3.6.2 of the paper (P18, L6 – 10) . We have modified the statement to make it more concrete (P1, L19).

p. l.17. What is meant by high wind speed. Which range are you referring to?

In this study, apart from the different inter-turbine spacings, we also explored the role of different wind speed ranges over which the wind turbines operate, ranging from lowest wind speed of 3 ms$^{-1}$ to highest wind speed of 18 ms$^{-1}$. Case A corresponds to low wind speed range and case C represents the high wind speed range. We have added this quantitative information on wind speed to the abstract (P1, L14–16).

p.2. l.15: What version of the WRF model? Which wind turbine parameterization ? Please rephrase the sentence in the abstract or in the introduction as they are identical.

We used WRF version 4.2.1 and the Fitch et al. (2012) wind farm parameterization for our simulations. We have added the WRF version (P3, L3). The wind turbine parameterization used is already mentioned in the paper (P3, L14). We have also rephrased the sentences in the introduction to avoid any duplication (P1, L29 – P2, L5).

p.4. l14ff: I suggest to add a figure showing the relation between grid cell and turbine spacing in order to makes things more clear for the reader

We have added a figure (Fig. 2a) showing the locations of turbines within the grid cells to make the relation between grid cell and turbine spacing clearer to the readers.

p. 5 l.15: This is still a simulation, so the term 'realistic' is not appropriate.

We have rephrased and rearranged the paragraph. The word "realistic" has been removed. (P6, L2).

p.6. l.5: Please introduce here what k,i,j is referring to.

The letters k, i, and j are the location indices in the vertical, zonal and meridional directions. We have added this explanation (P6, L15 – 16).

p.6. l.18:Over which domain are the horizontally averaged? Over the wind farm domain?

The horizontal averaging is over whole of domain 3 from the WRF simulations. We have added this explanation in the revised manuscript (P6, L15).

Eq. 8 Please describe what is defined by î and Äµ ?

The symbols î and ĵ are the unit vectors in the zonal and meridional directions, respectively. We have added this in the manuscript (P7, L10 – 11). We are unable understand the second symbol. We assume that it is a typo and the reviewer means j.

p.9. l15: Please give a broader description about the statistical analysis.

We have added a description (P10, L16 – P11, L3) and reference (P11, L1) about the Wilcoxon sign rank test.

Eq. 15: The denominator on the right-hand side of Eq. 9 should contain the unit. I also suggest to write 140m – 28m

Adding units to only one term in the equation will not be appropriate. We think the way we have written the equation with m in the numerator is causing some confusion. We have rewritten this equation using symbols to eliminate any confusion about units (P7, L24).

p.7 l.25 Why small v?

Sorry about the typo. We have corrected it in the manuscript (P8, L10).

Fig. 3. The figure is hard to read, especially 3c). Please enlarge the plots and the labels. Please add a unit to the scales. For case A the resolution is way to small to be able to follow the analysis on page. 9. l. 15-16.

As mentioned earlier, enlarging the figures will make it impossible to fit them in the same panel making it hard to compare between cases. We have replotted the figure with a RAINBOW type colour scheme for better visibility and added units to the labels.

p.10. l.5: Not true several other studies such as Platis et al. 2018, 2020, Siedersleben et al. 2018 reported a deceleration of up to 40 % in the wake of offshore wind farms.

The values reported in our manuscript are averages. The studies mentioned by the reviewers report maximum deceleration, e.g., Platis et al., 2018, 2020 found a maximum deceleration of 40% and 43% in the wake of offshore wind farms, respectively. However, we could not find any explicit mention of deceleration rates in Siedersleben et al., 2018. We have added this comparison in the text (P11, L20 – 23). We have added the references for Platis et al., 2018, 2020 in the revised manuscript.

p.10 l.25. What is the height of the ABL top?

The ABL top is ~1400 m for Case A and ~1000 m for Case B & C. We have added a line showing the ABL top height in Fig. 4.

Fig. 4. I recommend to mark the area where the wind farm is located.

The wind farm was already marked with a black dashed box in Fig. 4. It is also mentioned in the figure caption. We have made the dashed box that depict the wind farm cross-section bolder for better visibility in Fig. 4.

Fig. 4: Case A III. Why is there a deceleration and then an acceleration of the flow between 0-1000 m and  at x= 780-850 km?

Please note that the direction of wind flow is from right to left in Case AIII. Hence, the regions with red colour between 0-1000 m and at x= 780-850 km indicate deceleration upwind of the farm as seen in all cases. In this particular case, there is an alternating band of white and red. This pattern is perhaps giving the impression that there are alternating bands of acceleration and deceleration. However, the white coloured regions do not indicate acceleration. Rather, they denote regions where the signals are not statistically significant. Thus, this pattern actually indicates that the upwind deceleration in case AIII is relatively weak. We have added the explanation of the white patches in the figure caption (Fig. 4)

Fig. 4 The upwind deceleration seem very impressive. I am wondering whether a too small simulation domain is causing an intensification by boundary reflections? Is there a way to assess this influence ?

The domain is very large, 1500 km X 1500 km with a 50 km X 50 km wind farm in centre. The acceleration is not caused by boundary reflections. We had conducted sensitivity studies with domains of different sizes starting from a 300 km X 300 km domain. We found some wake reflections in the

smaller domains but there were no boundary effects with this large domain. We have added this explanation in Section 2.2 of the paper (P3, L22–24).

Fig. 4 b. Why is a streak pattern visible? Can this be also attributed to artifacts caused by the simulation?

Streak patterns are updrafts and downdrafts induced due to wind farm. We conducted significance tests on the results and plotted only the statistically significant signal to minimize the depiction of random noise.

p.11.9 ff. I do not understand how this argument contributes to case C-I. Please describe more clearly.

We assume the reviewer is talking about p11 L8 because P11 L9 talks about case III. We think this sentence is superfluous. We have removed this so as to no cause confusion (P12, L22).

Fig. 5 I do not understand the meaning of the legend at the second left figure in the first row.

These refer to the synoptic, mesoscale and microscale fluxes calculated using the equations described in Section 2.3.1. We have clarified this in the figure caption (Fig. 5).

Fig .5.  The description and argumentation of the results is at some points not very precise.

We have expanded Section 3.5 to improve our explanation of the pattern of vertical synoptic, mesoscale and microscale fluxes in a wind farm (P13, L4 – L5, L7, L14 – 15).

p.11. l.17.  It would be helpful for the reader just to mention again very briefly the difference between synoptic and micro scale.

We have added a brief description of the scales and referred to the appropriate equations in the text (P13, L4 – L5).

p.11 l.8-10. "It is possible that this meso-scale momentum transport aids in the wind farm recovery by making more momentum available for downward mixing by turbulence." This is very speculative. Is there a way to justify it ? Can the variations at different heights be explained?

We agree with the reviewer that this statement appears speculative. Indeed, that is why we have intentionally used the word 'possible' because the evidence for this statement is limited to only case C-I. In Case C-I, the UW mesoscale flux is maximum at ~1500 m altitude showing that the momentum is transferred from above free atmosphere (above the PBL height of ~1000 m) into the boundary layer. In Case C-I, a negative UW mesoscale flux at 1500 m height depict a net downward transport of higher momentum. The mesoscale flux is negligible up to 1000 m (around PBL height in the wind farm) because within the wind farm no vertical mesoscale momentum transport leads to downward transport

of momentum rather the downward transport of momentum happens through microscale fluxes. We have expanded this entire section to further clarify our point (P13, L14 – 16). Moreover, we have also provided detailed explanation of the microscale flux patterns (P14, L23 – L31).

Fig. 6: Why does the plots in case of III look much coarser than for I and II ?

The reviewer is absolutely right. In case III, we have a turbine in every other grid cell to ensure a 2 km spacing. The data points in this case are averaged over 4 grid cells covering 2 km X 2 km. That is why case III looks smoother. We have added this explanation to the text (P15, L14 – 16).

Fig. 7a) Why is there such a sharp boundary (jump) at about x= 4x10-3 ? Because of the different cases? This could be mentioned in the text.

Yes, the jump at x= $4x10^{-3}$ is for the different cases. We have added this in text (P17, L8 – 11).

p. 17 l. 11. Please refer to the equation or describe the integration by an separate equation.

We noticed there is a mistake in the text, it should have been section 2.3.1 instead of 2.3.3. As suggested by the reviewer, we have added the equation number for the reader's reference (P19, L7).

p.18 l. 31. What do you mean exactly by synoptic scale effects?

What we meant to say here is that wind farms do not affect any synoptic scale fluxes as evident from section 3.5, Fig. 5. In the mentioned figure, the effect on synoptic scale fluxes by the wind farm is negligible. This is the reason the synoptic scale flux line is not visible in the figure. We have stated this clearly in the conclusions (P23, L 9).

**Technical corrections**

p.2. l.1 ff: Please put the citations in chronological order: Done (P2, L3 – 5).

p.4 l.9: Please correct: boundary-layer scheme: Corrected (P4, L5).

p.4 l.10: Please correct: second-order moments: Corrected (P4, L6).

p.6 l.22 typo error 'that': Corrected (P7, L6).

Fig. 7 b)-d) Please title the figures with its specific cases. : The specific cases are added in the figure as titles (Fig. 7).

References:

Djath, B., Schulz-Stellenfleth, J., and Cañadillas, B.: Impact of atmospheric stability on X-band and C-band synthetic aperture radar imageryof offshore windpark wakes, Journal of Renewable and Sustainable Energy, 10, 043 301, 2018

Cañadillas, B., Foreman, R., Barth, V., Platis, A., Siedersleben, S. K., Bange, J., Lampert, A., Bärfuss, K., Hankers, R., Schulz-Stellenfleth, J.,Djath, B., Emeis, S., and Neumann, T.: Offshore wind farm wake recovery: Airborne measurements and its representation in engineeringmodels, Wind Energy, https://doi.org/10.1002/we.2484, 2019.

Platis, A., Hundhausen, M., Siedersleben, S. K., Lampert, A., Bärfuss, K., Schulz-Stellenfleth, J., Djath, B., Emeis, S., Neumann, T., Cañadil-385las, B., and Bange, J.: Long-range modifications of the wind field by offshore wind parks - results of the project WIPAFF, MetZet, 2020b

Siedersleben, S. K., Lundquist, J. K., Platis, A., Bange, J., Bärfuss, K., Lampert, A., Cañadillas, B., Neumann, T., and Emeis, S.: Micrometeorological impacts of offshore wind farms as seen in observations and simulations, Environmental Research Letters, 2018
* * *
**Reviewer 2 – WES-2021-7**

This study explores different recovery processes in a large hypothetical offshore wind farm in the Arabian Sea for 3 different meteorological conditions using the mesoscale model WRF and the Fitch WFP. The manuscript is well written in general. Some figures could be improved to faciliate the understanding of the reader. I think it is an interesting study that should potentially be published in WES, after carefully addressing the comments below.

**General comments**

My main criticism of this study is that, while the three different cases certainly consider different wind speed ranges, they might also have different stability ranges. However, the stability for the cases is not discussed (see specific comments below), although various studies have shown an impact of stability on wakes (Lee and Lundquist 2017, Cañadillas et al. (2020) to mention just a few).

This is a very important point. While designing the experiments to study recovery, we wanted to look at wind farm design, specifically, the variability of recovery process under different wind farm spacing and wind speeds. But we agree with the reviewer that the stability question is extremely important. As suggested by the reviewer, we have taken an extensive look at the static and dynamic stability patterns of the 3 case studies using lapse rate and Richardson number. We have added a section 3.2 (P10, L1 –L14) in the revised manuscript describing the stability conditions. We have also added section 3.8 (P19, L4 – L21) showing effect of stability on vertical recovery. We have also discussed the wake dependence on atmospheric stability in the revised manuscript (P11, L10 – L13). In the discussion

section, we have addressed the challenges in doing such simulations under realistic boundary conditions (P23, L33 – P24, L6).

My second concern is the sensitivity of the model results with respect to the vertical resolution. The vertical resolution is rather coarse (>20 m) compared to the resolution, which is suggested to be necessary to capture wind farm effects (e.g. Siedersleben et al 2020, Tomaszewski and Lundquist (2020) or Pryor et al. (2020)). In addition Pryor et al. (2020) also pointed out that the TKE magnitude depends on the vertical resolution. One of your conclusions concerns the effect of TKE advection on the recovery process. This conclusions could be faulty, if the resolution is too coarse. Please elaborate.

Thank you for bringing these studies to our notice.

Siedersleben et al. (2020) recommended a vertical level on the order of ~12m. However, they also mentioned that in case of limited computational resources, a horizontal resolution of 5 km and vertical resolution of 35 m below 100 m also captures the most important features over the wind farm. They looked at airborne measurements to evaluate the wind farm parameterization choices in WRF model for an offshore wind farm. They carried out sensitivity analysis of WF parameterization choices, such as TKE source, TKE advection and model resolution on TKE above the wind farms (around 60 m).

Pryor et al. (2020) carried out a sensitivity analysis of different modelling parameters in WRF for an onshore wind farm. They studied the role of wind farm parameterization schemes, vertical and horizontal resolution on the model outputs, such as, wind speeds, capacity factor, wake profiles, and TKE. Pryor et al. (2020) showed an impact of vertical resolution on TKE added in the wind farm and its vertical distribution.

Tomaszewski and Lundquist (2020) suggested a low-level vertical resolution of ~10 m and horizontal of either 1 km or 3 km in order to achieve the appropriate mixing required to match the expected surface warming and drying. They compared the WRF model results with observation data for an on-shore wind farm. In their study, the sensitivity analysis of WRF model resolution on hub-height wind speed deficits, near surface temperature and moisture changes was carried out .

Siedersleben et al. (2020) had suggested that a vertical resolution of 35 m below 100 m in case of limited computational resources also captures the most important features including TKE over the wind farm. Our simulations are computationally very expensive with each simulation costing ~16000 computational hours on a high-performance computer with 200 processors. We have conducted 24 simulations leading to a total of 384000 computing hours. Each simulation requires extensively large amount of computation resources because we have a very large 1500 km X 1500 km horizontal domain discretised with a high horizontal grid spacing (1 km for the finest grid). This configuration is essential for our simulations. We need the large horizontal domain to avoid wake boundary reflections and the fine horizontal resolution to capture the intra-wind farm wake effects for different inter-turbine spacing. We have used a vertical resolution of ~21.4 m in the lowest 150 m of model domain. This vertical resolution is also comparable to Pryor et al., 2020 (~16.7 m).

Lastly, we have used total 61 vertical levels till top of the atmosphere. It is necessary to keep a fine resolution beyond the rotor depth (lowest 150 m) as well because it will help us capture the effect of windfarm on mesoscale and synoptic fluxes also.

**Specific comments**

p 1, line 27ff: Is the citation of so many studies necessary here?

We have tried to provide an exhaustive list of studies that have explicitly discussed how recovery happens in the wind farms through microscale and mesoscale processes. Some of these studies have mentioned the recovery processes in passing while others have explored it using rigorous quantitative approaches. We have reduced the number of citations to more quantitative studies (P2, L3 – 5).

P 2, line 8: "… also confirmed by LES…", in line 5 you write that the recovery process was previously investigated by LES simulations. Why is the study by Calaf et al. (2010) listed separately from the others?

Sorry for missing the reference of Calaf et al. (2010) in with other citations. We have added the mentioned reference with others (P2, L9).

P 2, line 27: Here you write that TKE advection is deactivated in your study, while later on Page 5, line 14-17 you write that both activated and deactivated scenarios are used. Please make these statements consistent.

We agree that this sentence should be rephrased for clarity. We conducted two sets of simulations: the first set with TKE advection 'on' and the second with TKE advection 'off'. All results shown in sections 3.1 to 3.8 include TKE advection. In order to study the effect of TKE advection like Archer et al. (2020), we conducted sensitivity simulations with the TKE advection switched 'off'. The comparison of these two sets of simulations is discussed in section 3.9. We have made the mentioned statements consistent. We have also made the description clearer (P2, L31, P6, L1 – 5).

P 3, line 3: Please provide the version number of the WRF version that you have used. The bug found by Archer et al. (2020) showed, how important it is to document clearly, which version number has been used.

We have used WRF Vr 4.2.1 for our simulations. We have added this in the updated manuscript (P3, L3).

P3, line 19: How transferable are your results to other regions? This farm is quite far offshore. Existing offshore farms are closer to the coast. How could coastal effects change your results?

The results are valid for all deep offshore regions if the basic meteorology is same. However, in case of some extreme phenomena such as cyclones, we anticipate that results may vary. Coastal wind farms

are often affected by sea breezes and can have different recovery patterns. We are currently conducting a study for coastal wind farms but we do not have final results yet.

P3, line 24: A vertical resolution of > 20 m is at the upper limit of the necessary resolution to capture wind farm effects correctly as studies by Siedersleben et al 2020, Tomaszewski and Lundquist (2020) or Pryor et al. (2020) have indicated. In addition Pryor et al. (2020) also pointed out that the TKE magnitude depends on the vertical resolution. How sensitive are your results to vertical resolution?

Please see the justification given above in the general comments section.

P4, line 4: Was the Sea Surface Temperature also taken from that source?

Yes, the SST was also taken from the National Centers for Environmental Prediction Final Operational Global Analyses dataset (NCEP, 2015). We have updated this in the revised manuscript (P4, L3).

P4, line 4-11: A tabular overview could provide a better overview on the set-up

We have provided the physics parameterization that are used in the simulations in a tabular form in revised manuscript. (Table 1, P4).

P4, line 14: This is a rather small turbine compared to turbines that are currently being installed offshore. Why have you used such a small turbine and how representative are your results for more modern turbines?

We understand the reviewer's concern. The absolute magnitude of the power generation and recovery will depend on the turbine rating. However, we expect that the spatial patterns and relative magnitudes of the recovery will be valid for any of the installed turbine size provided that the inter-turbine distances and meteorological conditions are comparable. Please note that we have presented the recovery terms as a function of power generated in Figure 6 and Table 4.

P5, line 7: Atmospheric stability is another relevant parameter for wind energy. Is the stability similar in all three cases? Otherwise this will affect the conclusions as well.

We understand the reviewer's concern. We have analysed the stability conditions for all the three cases and added in the revised manuscript. We have added a section 3.2 (P10, L1 – L14) in the revised manuscript describing the stability conditions. We have also added section 3.8 (P19, L4 – L21) showing effect of stability on vertical recovery. In the discussion section, we have addressed the challenges in doing such simulations under realistic boundary conditions (P23, L33 – P24, L6).

P 6, Eq 6: Define what hat^i and hat^j mean. Consider to add subindex "h" to indicate that hub-height winds are used here.

The symbols î and ĵ are the unit vectors in the zonal and meridional directions, respectively. We have added this in the manuscript (P7, L10 – 11). We have added a sub-index "h" in Eq. (6) & (7) to indicate the hub-height wind speeds in order to avoid any confusions. (P7, L9 and L13).

P 8, line 18: How do these values compare with Volker et al. (2017)

We cannot directly compare Volker et al. (2017) with our study because the wind farm size and number of wind turbines are different than ours. However, the central tendencies of wind speeds are somewhat comparable. Here we present some very approximate comparisons by visual interpolation:

1) In our case, we are getting 31% efficiency for 4.5D and 11.3 m/s mean CTRL case wind speed. In case of Volker et al. 47.7% (interpolated) efficiency for 5.25 D and at a wind speed of 9.1 m/s.
2) In our case, we are getting 81% efficiency for 4.5D and 15.6 m/s mean CTRL case wind speed. In case of Volker et al. 60.3% (interpolated) efficiency for 5.25 D and at a wind speed of 13.1 m/s.
3) In our case, we are getting 78% efficiency for 8.9D and 11.3 m/s mean CTRL case wind speed. In case of Volker et al. 77.5% (interpolated) efficiency for 10.5 D and at a wind speed of 9.1 m/s.
4) In our case, we are getting 100% efficiency for 8.9D and 15.6 m/s mean CTRL case wind speed. In case of Volker et al. 85% (interpolated) efficiency for 10.5 D and at a wind speed of 13.1 m/s.

These comparisons are too approximate to be presented in the paper.

Figure 3, 4, 6: Please add variables and units to the colorbar.

We have added variables and units to the color-bars in the mentioned figures.

Figure 3: Are averages over the entire two or three day period shown in that figure? Or is it a snapshot? Please clarify. Consider to add the "plots depict the wind farm only, that is the black square in (c), if appropriate". In figure (c), please add: every xth vector is shown

In Fig. 3 (b) and (c), averages over the entire two day period is shown. This information has been added in the manuscript (P9, L10 and P10, L18). Rest of the suggested details have been added in Fig. 3 caption.

P 9, line 13: here it says averaged over the rotor depth, while in the caption of figure 3 "hub-height wind" is mentioned. Please clarify.

We apologize for the mistake. The plot shows differences in wind speeds averaged over the rotor depth, not the hub-height wind. We have corrected the error in the Fig. 3 caption.

P 9, line 17: The wake length depends also strongly on stability. Again, have you checked that the stability is the same in all simulations?

We have analysed the stability for the different cases and discussed this in the revised manuscript (P11, L10 – L13).

P 10, line 3: Wake lengths can be defined in many ways. You mention a wake length of 521 km, which is very long. Please add your definition of a wake length.

We take the difference of the hub-height wind speeds between the control and wind farm cases and estimate the statistical significance using the Wilcoxon sign rank test. The wake lengths are calculated as the distance from the wind farm to the grid cell where the difference falls below the 99% level of significance. We have added this in the manuscript (P11, L5 – 6).

P 10, line 13 – 15 and Figure 4: The figure could be easier understandable, if Case A would be turned into the mean wind direction and thus the colours would be the same in all cases. Please consider doing so.

We understand that this could be confusing for the readers. But we prefer to keep the figure axes unchanged because we want to give the readers a flavor of the fact that these simulations are constrained by observed boundary conditions where the wind directions vary. We have taken the components of meteorological variables along the dominant direction. To help readers understand these figures better, we have added arrows showing the prominent wind directions. We have made these arrows bolder so that the difference in wind direction between the different cases clearly discernible. (Fig. 4)

Figure 7 (a): Most points follow the linear trend. However there are also quite some points with relatively high momentum loss rate but small vertical recovery (especially around 10*10⁻³ ms⁻²). During which situations does this behaviour occur? Also there is a sharp drop at about 3*10⁻³ ms⁻², where does this come from?

There are also some points where though the momentum loss rate is high, the vertical recovery is low. These are the points on the upwind edges of the wind farm where the horizontal recovery dominates leading to lower values of vertical recovery. The sharp change at momentum loss rate ~ $3.5X10^{-3}$ in Fig. 7 (a), is because of change in cases. We have explained this in the revised manuscript (P17, L8 – 11).

Figure 8: Why is only the U-momentum flux considered here, even though the wind direction is south-west / north-east?

Figure 8 presented here is for Case C-I. This is the only case where the mesoscale fluxes are of the same order of magnitude as turbulent fluxes. The wind direction for case C is west-south-westerly with stronger u components than v. This is the reason the u momentum flux is much higher than the v momentum flux and hence, only the u-momentum flux is considered. We have added this explanation in the section 3.7 (P19, L2 – 4).

Section 3.7: I think this section deserves a bit more elaboration and possibly a figure. As you mention in your introduction on page 2, there has been a debate, whether TKE advection needs to be activated or not. With that in mind, it would be good to give more evidence for your conclusion on TKE advection.

We have added a table in the revised section 3.9 with values of difference in horizontal and recovery for the different cases analysed. We have also add a figure (Fig. 11) showing spatial patterns of vertical and horizontal recovery with TKE advection 'off'.

**Technical corrections**

P5, line 21: Consider to add "the method by" in using Avissar and Chen (1993).

We have added "the method by" (P7, L1).

P6, line 22: "that" is used before and after the citation.

We have removed the repetition (P7, L6).

P8, line 7: "As" should not be capitalized.

Sorry for typo, we have corrected it (P8, L17).

P8, line 23-24: Fig. 3b is referenced twice within one sentence

We have corrected it (P9, L11).

Figure 3: sub-figures are usually referenced only with (b) not with Fig. 3 (b). Please check with the journal guidelines

We have revised the figure caption (Fig. 3)

P 13, line 22: consider to add "side" or the like after "at the upwind"

We have added "side" after upwind (P15, L9).

P 18, line 28: "till" is rather colloquial. Consider to use "until" instead of "till"

We have replaced "till" by "until" (P23, L6).

P 19, line 24 – 28: I agree with the authors that it is an interesting study, but how significant a contribution is cannot be known before publishing. I would suggest to advertise it a bit less.

We have rewritten this paragraph as (P24, L 7).

P 19, line 31: Code availability: Please add the URL to the WRF github repository (or where ever you downloaded WRF)

We had downloaded WRF version 4.2.1 from github. We have added a URL ([https://github.com/wrf-model/WRF/releases/tag/v4.2.1](https://github.com/wrf-model/WRF/releases/tag/v4.2.1)) for the same (P24, L13).

P 19, line 32: Code availability: Please also add namelist.wps to the the repository. Why are only the example files for 0.5 km included in the repository, even if they are modified within the code it would be good to have them all. Please also add a description of model changes in the 5 files that you mention in the README.

We have updated the repository as per the suggestions. The new doi for the same is updated in the manuscript (P24, L14).

P 20, line 1-2: Please add a link to the NCEP data.

We have added the doi (P24, L18).

P 21, line 10: Chou et al: How can this publication be accessed? Please provide a doi / url

We have added the url.

P 21, line 30-31: Hong et al: How can this publication be accessed? Please provide a doi / url

We have added the url.

P 22, line 22: line 1-2: IRENA: How can this publication be accessed? Please add a url.

We have added the url.

P. 22, line 7: Li et al: How can this publication be accessed? Please provide a doi / url

We have added the doi.

P. 22, line 16: Noppel et al: How can this publication be accessed? Please provide a doi / url

We have added the doi.

P 22, line 26: Skamarock et al: How can this publication be accessed? Please provide a doi / url. In addition: your zenodo repository indicates that you have been using WRF 4.2.1, why do you cite version 3 here?

We have corrected the citation for WRF 4.2.1.

P. 23 line 5: Zeng et al: How can this publication be accessed? Please provide a doi / url

We have added the doi.

**References**

Cañadillas B, Foreman R, Barth V, Siedersleben S, Lampert A, Platis A, Djath B, Schulz-Stellenfleth J, Bange J, Emeis S, Neumann T (2020) Offshore wind farm wake recovery: Airborne measurements and its representation in engineering models. Wind Energy 23(5):1249–1265, DOI 10.1002/we.2484

Lee JCY, Lundquist JK (2017b) Observing and Simulating Wind-Turbine Wakes During the Evening Transition. Boundary-Layer Meteorology 164(3):449–474, DOI 10.1007/s10546-017-0257-y

Siedersleben, S.K., Platis, A., Lundquist, J.K., Djath, B., Lampert, A., Bärfuss, K., Cañadillas, B., Schulz-Stellenfleth, J., Bange, J., Neumann, T. and Emeis, S.: Turbulent kinetic energy over large offshore wind farms observed and simulated by the mesoscale model WRF (3.8.1), Geosci. Model Dev. (Online), 13(NREL-JA-5000-75874), https://doi.org/10.5194/gmd-13-249-2020, 2020.

Tomaszewski JM, Lundquist JK (2019) Simulated wind farm wake sensitivity to configuration choices in the Weather Research and Forecasting model version 3.8.1. Geoscientific Model Development Discussions, Final version published after completion of this study at DOI 105194/gmd-13-2645-2020 DOI 10.5194/gmd-2019-302

Pryor SC, Shepherd TJ, Volker PJH, Hahmann AN, Barthelmie RJ (2020) "Wind Theft" from Onshore Wind Turbine Arrays: Sensitivity to Wind Farm Parameterization and Resolution. Journal of Applied Meteorology and Climatology 59(1):153–174, DOI 10.1175/jamc-d-19-0235.1

Volker PJH, Hahmann AN, Badger J, Jørgensen HE (2017) Prospects for generating electricity by large onshore and offshore wind farms. Environmental Research Letters 12(3), DOI 10.1088/1748-9326/aa5d86

---

## Author Response (AR2)

**Report 1: (Referee 2)**

I would like to thank the authors for their work on improving the manuscript. I have a few more comments. Note that the page and line numbers refer to the version with tracked changes

We thank the reviewer for the encouraging comments. The reviewer's comments are marked in blue. We provide below a point-by-point response to the reviewer's comments. The page numbers and line numbers refer to the version with tracked changes.

**Specific comments:**

1. P 17, line 9: I don't understand "change in cases", which "cases"? A, B and C?

   Vertical recovery increases with increasing momentum loss rate. The increase in vertical recovery is very sharp at the transition from case II to case I at momentum loss rate ~ $3.5 \times 10^{-3}$ in Fig. 7a. We have added this explanation in the revised manuscript on P18, L22.

2. P 21, Figure 10: Why does this figure contain much fewer points than Figure 7? Is this due to the use of hourly values? Then this should be mentioned in the caption.
   It is a bit difficult to see whether there is no relationship, since the points are plotted on top of each other. Maybe, an x-axis with discrete stability classes could be used instead of wind speed (similar to figure 9).

   Figure 7 plots all the 50x50 points over the wind farm for 48 hours. Thus, the total number of data points in Fig. 7a are 9x50x50x48 and in Fig. 7b-d are 3x50x50x48 each. In contrast, Fig. 9j-l and Fig. 10j-l (updated) plots the vertical recovery spatially averaged over all the 50x50 points of the wind farm for 48 hours. Thus, the total number of data points in Fig 9j-l. and Fig. 10j-l (updated) are only 3x48 for each sub-plot. We have added this information to the captions of Fig. 7 and Fig. 9, and Fig. 10 to avoid all possible confusion.

   We agree with the reviewer that Figs 9 and 10 can cause confusion. We have remade Figs. 9 and 10. Fig. 9 a-i shows the relationship between vertical recovery and $R_f$ for different wind speed bins. Fig. 9 j-l shows the relationship between vertical recovery and wind speed for the 3 stability bins. Here stability is estimated from $R_f$. The figure demonstrates that vertical recovery does not have a strong relationship with stability in our experiments but it does have a strong relationship with wind speed. Very similar patterns are visible in Fig.10 that is exactly the same as Fig. 9 but here stability is estimated using the non-local method instead of $R_f$. These two figures show that the variation in vertical recovery is dominated by wind speed and not by stability for these set of simulations. We have clarified this in the revised manuscript (P21, L9 to P22, L9). However, we understand that our simulations are not comprehensive enough to identify any definitive relationship between stability with vertical recovery. This limitation has been explained in Section 4.

3. P 21, line 6: I would call the effect of TKE on recovery not "minimal", since you estimated it to be 5 %. Maybe "small" would be more appropriate?

   We agree with the reviewer and have changed 'minimal' to 'small' (P23, L2; P24, L31).

4. P 21, line 11: I think for the horizontal recovery the spatial pattern looks different for B+C. With advection turned on, the pattern was more chaotic and decreasing downstream.

Having it turned off, these alternating patterns of positive and negative bands emerge more.

When discussing the spatial pattern of recovery, we meant to say that horizontal recovery is strong at the upwind edges and vertical recovery is strong in the interiors for both TKE advection 'on' and 'off' simulations. However, the reviewer is right in saying that the banded structure in horizontal recovery is more prominent in the TKE advection 'off' simulations. We have clarified this in the manuscript on P23, L5-7.

**Technical comments:**

1. p 10, line 2: approach → approaches; "," → "."
   Done. (P8 L15)

2. p 10, line 7: add "if"
   Done. (P9, L1)

3. p 15, figure 6/22, figure 11: I would suggest to use the same color scale in (b) for both figures, so they can be easier compared
   Done.

4. p 21, table 5 caption: consider to add "area depicted in Fig. 11" after "averaged over the wind farm, since it is only mentioned in figure 3b that this 50x50 km covers the entire wind farm
   We have modified the caption to:
   Table 5: Change (TKE advection 'off' – TKE advection 'on') in vertical recovery (x $10^{-3}$), ms$^{-2}$ and horizontal recovery (x $10^{-3}$), ms$^{-2}$, averaged over the 50x50 km$^2$ area of the wind farm, the 48-hour simulation period, and cases A, B & C. The numbers in the parenthesis give the percentage change in recovery with respect to the corresponding momentum loss rate. * denotes that the values are significant at $p<0.01$.
* * *
**Report 2: Referee 1**

The authors have adequately answered all my questions and performed the needed changes in their manuscript. I therefore recommend the the study for publication with subject to some minor revisions.

We thank the reviewer for the encouraging comments. The reviewer's comments are marked in blue. We provide below a point-by-point response to the reviewer's comments. The page numbers and line numbers refer to the version with tracked changes.

1. L.13ff Instead of giving a distance in km I would express the distance in term of rotor diameter or add this information.
   We have modified the abstract as follows (P1, L14):
   Different inter-turbine spacings range from a densely packed wind farm (Case I: low inter-turbine distance of 0.5 km ~5 rotor diameter) to a sparsely packed wind farm (Case III: high inter-turbine distance of 2 km ~20 rotor diameter)

2. L.23 language!

Sorry, we corrected this in the text but missed it in abstract. We have now edited the abstract as follows (P1, L23):
"To the best of our knowledge, this is one of the first studies to look at wind farm replenishment processes …."

3. Fig. 3 I personally find the figures still too small and would prefer to split them. I leave this decision to the editor.
We have enlarged Fig. 3c.

4. Chapter 3.2. Please introduce the equation of the Flux Richardson Number. Over how many heights have you calculated the fluxes and averaged them?
I am not sure if it makes sense to show Rf=1 since it is very unlikely that the value gets exactly 0. Have you though of introducing bins instead of distinct number to have a smoother transition between the different regimes?
We have added section 2.4 in the revised manuscript with the equation of the Flux Richardson Number ($R_f$). $R_f$ is averaged from the surface to the wind turbine rotor tip height of 140 m of the wind farm cases that comprises of around 7 eta levels. We have added this information in the revised manuscript (P11 L6)
We agree with the reviewer that instead of neutral, we should consider near-neutral stability. We have revised the Table 3 and Fig. 9 accordingly. We also added the description of the bins. We have rewritten the text in Section 2.4 (P8, L15-20) as follows:
"In the first approach, we calculate the Flux Richardson number ($R_f$, Stull, 2012) as per Eq. (11). As per Sorbjan and Grachev (2010), $R_f < -0.02$, $-0.02 < R_f < 0.02$, and $R_f > 0.02$ correspond to statically unstable, near-neutral and stable environments."

5. p.11 L1ff I am not sure if the method of the Wilcoxon Sign test is well-known in the wind eenrgy community. I personally suggest to give a broader description of the method as used with your data.
The test was chosen because the input data was not following the normal distribution. Wilcoxon sign rank test is a non-parametric test that does not require the data to follow a normal or other known distribution. Hence, this is a common alternative to other tests that need the data to follow a normal distribution. The results shown in Fig. 3c are for the points in the domain where we reject the null hypothesis and claim that the difference in the wind speeds (WF-CTRL) is not because of random chance at a confidence level of greater than 99%. We have added this information in the revised manuscript. (P12, L1-8)

6. Fig. 4 Why is the ABL top decreasing with downstream distance of the wind farm for case A?
No, the ABL top is increasing in case A with downstream distance of the wind farm. Please note that the direction of wind reverses in case A as compared to B & C.

7. p.14 L-23-31. Interesting! This would imply a subsidence of the whole air mass above the wind farm. Have you seen any "adiabatically" warming effect above the wind farm in your simulation ?
It will be very interesting to study if the subsidence causes any warming effect above the wind farm. However, we did not study the thermodynamic aspects of the wind farm-atmosphere interactions and limited the scope of this paper only to wind farm dynamics.

8. p.19 L 15. You can skip the reference here since it has already been introduced in the previous section.
Ok. We have removed the reference. (P20, L18)

9. Chapter 3.8 Have you checked for a relationship between recovery and lapse rate?

Yes we have checked the relationships between vertical recovery and lapse rate. We have added Fig. 10a-i to show the relationship between stability estimated based on non-local lapse rate and recovery. We did not find any evidence of a strong relationship.